# Learning to Recover Task Experts from a Multi-Task Merged Model

## Abstract

Multi-task model merging aims to merge several task-specific models (or experts) into a unified multi-task model. However, model merging often results in performance degradation due to parameter interference between experts. While several recent works have focused on improving the merging process to mitigate the parameter interference, there still exists the performance gap between merged models and task experts. In this work, we take a different perspective: we aim to recover a task expert from a merged model, instead of trying to improve the merging process. We first note that the parameter interference arises, as a merging process introduces offsets to expert model parameters. Thus, we propose to learn to **Re**cover a **T**ask e**X**pert (**ReTeX**) model, by undoing this offset. Specifically, we train a lightweight linear module to predict the offset to recover a task expert for a given input. Experiments demonstrate that ReTeX significantly outperforms existing model merging methods across computer vision domains and NLP domains with models of various scales, recovering more than 99% of individual expert performance even when scaling to 30 tasks. Furthermore, ReTeX can be applied to several existing merging models, demonstrating its flexibility and applicability.

## 1 Introduction

Ever since the advent of foundation models (Achiam et al., 2023; Saab et al., 2024; Ding et al., 2023) pre-trained on large-scale data, deep learning models have displayed striking success across various domains, through fine-tuning such large-scale pre-trained models on each task. However, the use of such task-specific models that are trained and stored independently raises a question: if they all share the same structure and same initialization (i.e., a pre-trained foundation model), can we integrate the knowledge from task-specific models into a single model?

Multi-task model merging (Ilharco et al., 2023; Yadav et al., 2023; Yang et al., 2024) has emerged as a promising solution. Multi-task model merging aims to integrate knowledge from various task experts through weighted summation of parameters of task experts, weighted by task coefficients that encode the importance of each parameter for each task. Early works have mainly focused on merging existing fine-tuned models into a single merged model, obtained via the weighted summation of the parameters with fixed universal task coefficients (Ilharco et al., 2023; Jin et al., 2023a;b; Matena, 2022; Yadav et al., 2023; Yang et al., 2024; Tang et al., 2023). However, these merging methods have struggled to find a single merged model that could perform as well on all tasks as corresponding task experts.

In light of the challenge, recent multi-task model merging methods have tried to dynamically find better merging coefficients for each input task during inference (Oh et al., 2025; Tang et al., 2024; Lu et al., 2024; Muqeeth et al., 2023). This input-adaptive merging scheme has led to substantial performance improvement, however with an increased memory overhead. These approaches methods require all task-specific model parameters or components in memory during inference, as a merged model is formed on the fly by combining task experts with input-adaptive coefficients. Yet, these merged models are still formed via merging that inherently introduces parameter interference, underperforming compared to task-specific experts on their respective tasks.

In this work, instead of trying to improve a merging process, we approach the problem from a different perspective: we aim to **Re**cover **T**ask e**X**perts from a merged model. From the perspective of task experts, a merging process inevitably introduces parameter interference, as parameter offsets

are introduced from other task experts during model merging. Thus, our key insight is that each expert model can be recovered from a merged model, if we can *undo* the offset introduced by a merging process. Building on this insight, we train a lightweight linear layer that learns to find the offset for each input. We use this offset to recover a task expert from a merged model.

We note that RETEX can be applied to both scenarios, when task distribution is known for each input (task-known scenarios, which is the standard multi-task model merging setting (Yadav et al., 2023; Ilharco et al., 2023; Huang et al., 2024)) or unknown for each input. For task-known scenarios, the offset prediction module in RETEX simply employs the task identity of each input. For task-unknown scenarios, RETEX employs an independently trained task-id router and uses the output of the router to estimate task identity.

We further note that the training of the offset prediction module in RETEX does not require training data nor test samples. Since the offset prediction just needs to learn the parameter offset from merged parameters to task-expert models for any given task identity, we can just randomly sample task identity. Then, for each sampled task identity, the corresponding task expert parameters will be used as ground-truth during the training of RETEX. Once the training of RETEX is finished, existing task expert models are no longer required during inference. This enables post-hoc deployment over existing task-specific models.

Through extensive experiments with several merged models and backbones of varying scale across both computer vision and natural language processing tasks, we demonstrate the efficacy, efficiency, and flexibility of RETEX. Particularly, RETEX recovers over 99% of individual-expert performance even when scaling to 30 tasks, without incurring large inference-time memory overhead compared to previous works.

## 2 RELATED WORKS

Multi-task model merging consolidates multiple fine-tuned models, often from a common pre-trained foundation, into a single network without retraining, aiming for efficient multi-task capability and reduced deployment overhead. Early approaches involved direct weight averaging (Utans, 1996; Shoemake, 1985; Ilharco et al., 2022), which often suffered from performance degradation due to task interference. More sophisticated static methods like Fisher-Merging (Matena, 2022) used parameter importance (via Fisher Information) for weighted combinations, while RegMean (Jin et al., 2023b) explored principled averaging with regularization, though task interference remained a key challenge. Task Arithmetic (Ilharco et al., 2023) offered a conceptual shift by introducing task vectors (parameter difference from a base model), allowing arithmetic combination of these compact representations. This spurred methods like TIES-Merging (Yadav et al., 2023), which manipulates task vectors (e.g., sparsification, sign resolution) to mitigate interference, and TALL-Mask (Wang et al., 2024b), which identifies salient task-specific parameters within a merged model by analyzing parameter differences to create task masks.

Building on task vector concepts (Ilharco et al., 2023), many studies reduce parameter interference by enforcing sparsity or operating in compact parameter regions (Deep et al., 2024; He et al., 2024; Wang et al., 2024b; Davari & Belilovsky, 2024; Zhu et al., 2024; Kong et al., 2024). DARE (Yu et al., 2024) drops low-magnitude updates and rescales salient weights, while AdaMerging (Yang et al., 2024) optimizes coefficients at model or layer granularity via test-time adaptation on evaluation data. EMR-Merging (Huang et al., 2024) maintains a shared backbone together with sparse, task-specific components by selecting dominant parameter values across tasks. However, several of these approaches require task-dependent hyperparameter tuning (e.g., TIES (Yadav et al., 2023), TALL-Mask (Wang et al., 2024b)) or swapping task-conditioned modules at inference (e.g., EMR-Merging (Huang et al., 2024)), which presupposes access to task identity and increases management overhead as the number of tasks grows.

A complementary line of work adjusts combining coefficients or activates specialized branches at inference based on the input (Kang et al., 2024; Li et al., 2023; Muqeeth et al., 2023; Lu et al., 2024; Tang et al., 2024; Oh et al., 2025). Examples include learned routers that mix expert subnet-works (Muqeeth et al., 2023; Lu et al., 2024) and schemes that compute coefficients from uncertainty or entropy without extra training signals (Oh et al., 2025). These techniques often achieve strong ac-

curacy, but they typically keep multiple expert checkpoints, masks, or routing modules available at run time and may require additional forward passes per sample, increasing memory use and latency.

By contrast, instead of trying to find better merging coefficients, we take a different approach: aiming to recover a task expert from a merged model. Upon our insight that merging process undermines the performance due to offsets introduced to task expert parameters, our method RETEX delivers task-expert-level performance by learning to *removing* such offset.

## 3 BACKGROUND

**Problem setting.** Given a pre-trained model $f : \mathcal{X} \times \Theta \to \mathcal{Y}$ with parameters $\boldsymbol{\theta}_0 \in \Theta$, we assume there are task-specific models, fine-tuned from the pre-trained model $f$ to each downstream task $t \in \{1, \dots, T\}$. In other words, we assume there are $T$ task-specific models $\{f_{\boldsymbol{\theta}_t}\}_{t=1}^T$, each with parameters $\boldsymbol{\theta}_t$ obtained by fine-tuning the pre-trained model on the corresponding dataset $\mathcal{D}^{(t)} = \{(\boldsymbol{x}_i^{(t)}, y_i^{(t)})\}_{i=1}^{N_t}$, where $\boldsymbol{x}_i^{(t)} \in X^{(t)} \subseteq \mathcal{X}$ is an input with a corresponding label $y_i^{(t)} \in Y^{(t)} \subseteq \mathcal{Y}$. The goal of multi-task model merging (Matena, 2022; Jin et al., 2023b) is to find the task coefficients $\{\alpha_t\}_{t=1}^T$ that would result in a merged model $\boldsymbol{\theta}_{\text{merged}} = \sum_{t=1}^T \alpha_t \boldsymbol{\theta}_t$ that can perform as well as each task-specific model on the respective task. Then, the merged model will perform prediction for each new input data $\boldsymbol{x}$, which can come from any task $t$. Under standard settings (Yadav et al., 2023; Ilharco et al., 2023; Huang et al., 2024), the task identity $t$ of $\boldsymbol{x}$ is assumed to be known (hence, task-known scenarios). Otherwise, under task-unknown scenario, the task identity $t$ is unknown. In this work, we tackle both scenarios.

**Task arithmetic.** To better facilitate the knowledge manipulation, Task Arithmetic (Ilharco et al., 2023) has introduced the concept of task vector. For each task-specific model $f_{\boldsymbol{\theta}_t}$, task vector $\boldsymbol{\tau}_t$ is obtained by subtracting the pre-trained model parameters $\boldsymbol{\theta}_0$ from task-specific model parameters $\boldsymbol{\theta}_t$. Hence, task vector $\boldsymbol{\tau}_t$ is a vector pointing towards $\boldsymbol{\theta}_t$ from $\boldsymbol{\theta}_0$, representing the task-specific knowledge for task $t$. Leveraging the task vector concept, subsequent works (Yadav et al., 2023; Wang et al., 2024b) have formulated the model merging process as

$$\boldsymbol{\theta}_{\text{merge}} = \boldsymbol{\theta}_0 + \sum_{t=1}^T \lambda_t \boldsymbol{\tau}_t, \tag{1}$$

where $\lambda_t$ represents task coefficients under task arithmetic scheme.

## 4 LEARNING TO RECOVER TASK EXPERTS

Previous merging methods have attempted to find merging coefficients that would provide better performance on each task. As such, recent works (Oh et al., 2025; Tang et al., 2024; Lu et al., 2024; Muqeeth et al., 2023) have tried to find input-adaptive merging coefficients for each input during inference. The input-adaptive merging process has lead to performance improvement, however at the cost of memory usage. Yet, there still exists the performance gap between merged models and the task experts.

We believe that the reason for the persisting gap is that merged model parameters are the shifted version of task expert parameters due to parameter offsets introduced during a merging process. Thus, in this work, we take a different perspective: instead of further optimizing a merging rule, we undo the interference of a given merged model by recovering each expert directly from it. Concretely, we posit that each task expert can be written as the merged parameters plus a task-specific offset

$$\boldsymbol{\theta}_t = \boldsymbol{\theta}_{\text{merge}} + \boldsymbol{\beta}_t, \tag{2}$$

where $\boldsymbol{\beta}_t$ is the offset that corrects the deviation of $\boldsymbol{\theta}_{\text{merge}}$ from the true expert $\boldsymbol{\theta}_t$. This offset view is a direct way to model (and remove) the interference introduced during merging; we provide justification and derivations for adopting the offset form in Appendix A.

The overall framework is illustrated in Figure 1-(a): given a predicted task ID, RETEX recovers the corresponding task expert from the merged parameters $\boldsymbol{\theta}_{\text{merge}}$ by estimating $\boldsymbol{\beta}_t$. During inference, we first determine the task ID for an input $\boldsymbol{x}$ (Section 4.2). As shown in Figure 1-(b), RETEX then generates a task-conditioned offset and adds it to $\boldsymbol{\theta}_{\text{merge}}$ to recover the expert parameters (Section 4.1).

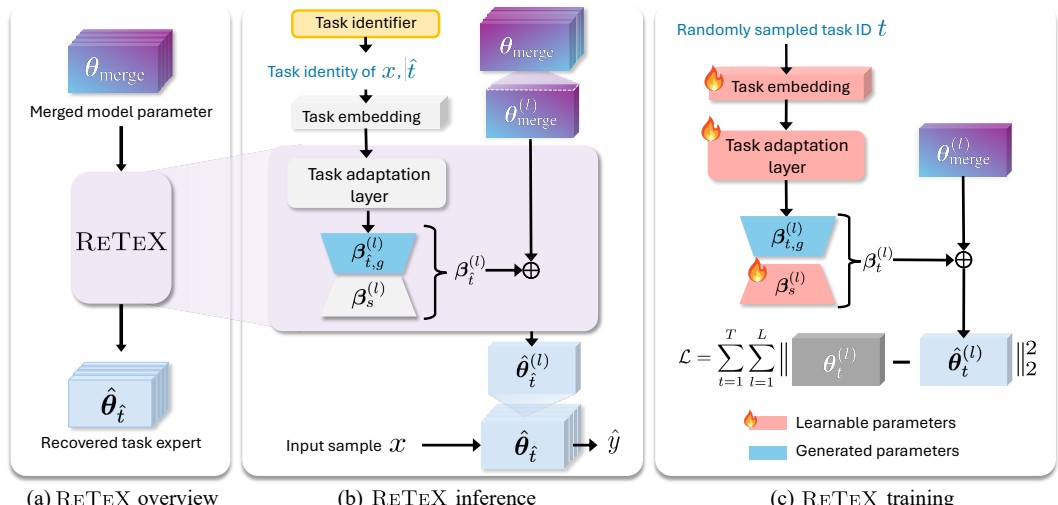

(a) RETEX overview      (b) RETEX inference      (c) RETEX training

Figure 1: **Overview of our proposed RETEX framework.** (a) RETEX overview: For given merged model parameters $\boldsymbol{\theta}_{\text{merge}}$, RETEX recovers the task expert $\hat{\boldsymbol{\theta}}_t$. (b) RETEX inference: An input $\boldsymbol{x}$ obtains its task ID from the task identifier, which is then mapped to a task embedding and fed to a lightweight task adaptation layer. For each layer $l$, the adaptation layer generates the low-rank factor $\boldsymbol{\beta}_{t,g}^{(l)}$, which combines with a shared learnable low-rank matrix $\boldsymbol{\beta}_s^{(l)}$ to form the layer offset $\boldsymbol{\beta}_t^{(l)} = \boldsymbol{\beta}_{t,g}^{(l)} \boldsymbol{\beta}_s^{(l)}$. Adding these offsets to $\boldsymbol{\theta}_{\text{merge}}$ yields the recovered task expert $\hat{\boldsymbol{\theta}}_t$. (c) RE-TEX training: A randomly sampled task ID $t$ is embedded and fed to a lightweight task adaptation layer. The learnable parameters are the task embedding, the adaptation layer weights, and $\boldsymbol{\beta}_s^{(l)}$, while the adaptation layer generates $\boldsymbol{\beta}_{t,g}^{(l)}$. $\boldsymbol{\beta}_{t,g}^{(l)}$ and $\boldsymbol{\beta}_s^{(l)}$ are combined with $\boldsymbol{\theta}_{\text{merge}}$ to recover the task expert parameters $\hat{\boldsymbol{\theta}}_t^{(l)}$. Training proceeds by minimizing the difference between the recovered parameters $\hat{\boldsymbol{\theta}}_t^{(l)}$ and the target parameters $\boldsymbol{\theta}_t^{(l)}$.

## 4.1 TASK-EXPERT RECOVERY

**Task embedding.** To generate the task-conditioned offsets $\boldsymbol{\beta}_t$, we represent each task $t$ with a learnable embedding $e_t \in \mathbb{R}^{d_{\text{emb}}}$, which captures task identity and conditions the offset generator.

**Layer-wise offset generation.** Motivated by layer-wise merging schemes (Yang et al., 2024) and efficiency, we parameterize the offset at each layer $l$ in a low-rank form. Let $\boldsymbol{\theta}_{\text{merge}}^{(l)} \in \mathbb{R}^{a \times b}$ denote the merged parameters and choose a rank $r < \min(a, b)$. Conditioned on the task embedding $e_{\hat{t}}$, a lightweight adaptation module $h^{(l)}$ produces a task-conditioned factor $\boldsymbol{\beta}_{\hat{t},g}^{(l)} \in \mathbb{R}^{a \times r}$. This is multiplied by a shared learnable factor $\boldsymbol{\beta}_s^{(l)} \in \mathbb{R}^{r \times b}$ (initialized at zero and shared across tasks) to form the layer offset:

$$\boldsymbol{\beta}_{\hat{t}}^{(l)} = \boldsymbol{\beta}_{\hat{t},g}^{(l)} \boldsymbol{\beta}_s^{(l)}. \tag{3}$$

The recovered expert parameters at layer $l$ are then

$$\hat{\boldsymbol{\theta}}_{\hat{t}}^{(l)} = \boldsymbol{\theta}_{\text{merge}}^{(l)} + \boldsymbol{\beta}_{\hat{t}}^{(l)}. \tag{4}$$

Stacking layers yields $\hat{\boldsymbol{\theta}}_{\hat{t}} = \boldsymbol{\theta}_{\text{merge}} + \boldsymbol{\beta}_{\hat{t}}$ as in Equation 2. This design keeps generation lightweight (only $\boldsymbol{\beta}_{\hat{t},g}^{(l)}$) while amortizing capacity through the shared $\boldsymbol{\beta}_s^{(l)}$.

**Training objective.** RETEX does not require training or test inputs. We train the offset generator using only task IDs: sample a task $t$, form $\hat{\boldsymbol{\theta}}_t^{(l)}$ by Equation 3–Equation 4, and minimize the L2 distance to the ground-truth expert parameters:

$$\mathcal{L} = \sum_{t=1}^{T} \sum_{l=1}^{L} \left\| \hat{\boldsymbol{\theta}}_t^{(l)} - \boldsymbol{\theta}_t^{(l)} \right\|_2^2. \tag{5}$$

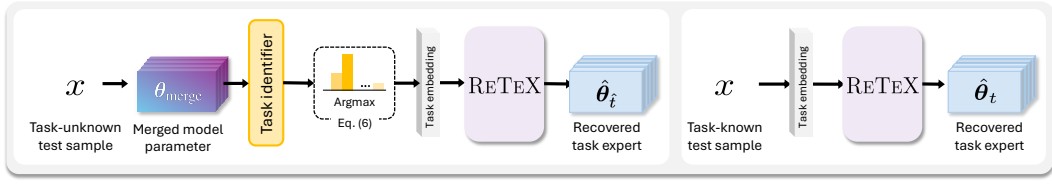

(a) Task-unknown scenario            (b) Task-known scenario

Figure 2: **Routing for task ID in RETEX.** RETEX inference with (a) *Task-unknown scenario* and (b) *Task-known scenario* inputs. When the task identity is unknown, a lightweight router predicts the task ID, which is then embedded and used in RETEX to recover the corresponding task expert. If the task identity is known, the given ID is directly used in RETEX to recover the expert.

After training, task experts $\{\boldsymbol{\theta}_t\}$ are no longer needed at inference; RETEX recovers experts on-the-fly from $\boldsymbol{\theta}_{\text{merge}}$ using the predicted task ID and the lightweight generators.

## 4.2 TASK CLASSIFICATION

To determine which task expert to recover for a task-unknown input $\boldsymbol{x}$, we first infer its task ID $\hat{t} \in \{1, \dots, T\}$ with a lightweight router $\mathcal{R}_\phi$. As illustrated in Figure 2-(a), the input $\boldsymbol{x}$ is forwarded through the merged model $\boldsymbol{\theta}_{\text{merge}}$ to obtain a final embedding $\boldsymbol{\theta}_{\text{merge}}(\boldsymbol{x})$, which the router maps to logits over $T$ tasks; the predicted task ID is

$$\hat{t} = \underset{t \in \{1, \dots, T\}}{\arg\max} \left[ \mathcal{R}_\phi\big(\boldsymbol{\theta}_{\text{merge}}(\boldsymbol{x})\big) \right]_t. \tag{6}$$

We train $\mathcal{R}_\phi$ with cross-entropy on task indices using a small, balanced calibration set, adding negligible overhead. When the task identity is known at inference, we skip the router entirely (Figure 2-(b)) and directly use the given task ID to retrieve the corresponding task embedding and generate the offsets in Section 4.1. In both cases, the (predicted or provided) task ID conditions the offset generator to recover the appropriate task expert from $\boldsymbol{\theta}_{\text{merge}}$.

## 5 EXPERIMENTS

In this section, we first study efficiency improvements of RETEX, and then evaluate its multi-task merging performance under two inference scenarios: *task-known* and *task-unknown*. In the *task-known* scenario (task identity available at inference), which is a standard multi-task model merging setting, we compare with methods that construct a merged model conditioned on a given task: Task Arithmetic (Ilharco et al., 2023), TIES-Merging (Yadav et al., 2023), and EMR-Merging (Huang et al., 2024). In the *task-unknown* scenario (task identity not given), we compare with methods that operate without task identity: Weight Averaging, Twin-Merging (Lu et al., 2024), and DaWin (Oh et al., 2025). RETEX supports both settings; when task identity is unknown, we predict it with the lightweight router (Sec. 4.2) and recover the corresponding expert, whereas when task identity is known, we directly recover the specified expert.

**Training setup.** Unless specified otherwise, the base merged model $\boldsymbol{\theta}_{\text{merge}}$ upon which RETEX operates is constructed using simple Weight Averaging of the task-specific expert models; this choice is made to demonstrate the capability of RETEX to recover task experts even from a minimally complex, conventionally defined merged model. We train RETEX for 5000 iterations. The optimization is performed using the Adam optimizer (Kingma, 2015) with an initial learning rate of $2 \times 10^{-4}$. The learning rate schedule follows a cosine annealing approach, incorporating 600 warm-up steps. The objective function for training RETEX is the L2 loss between the reconstructed layer parameters $\hat{\boldsymbol{\theta}}_t^{(l)}$ and the task expert layer parameters $\boldsymbol{\theta}_t^{(l)}$, as defined in Equation 5, summed over all tasks and layers.

Table 1: **Multi-task performance of merged models across different CLIP backbones and numbers of tasks.** Values in parentheses $_{(.)}$ indicate normalized accuracy (merged / individual). All methods are evaluated on 8, 14, and 20 computer vision tasks.

| Method | ViT-B/32 | | | ViT-B/16 | | | ViT-L/14 | | |
|---|---|---|---|---|---|---|---|---|---|
| | 8 tasks | 14 tasks | 20 tasks | 8 tasks | 14 tasks | 20 tasks | 8 tasks | 14 tasks | 20 tasks |
| Zero-shot | 48.3 | 57.2 | 56.1 | 55.3 | 61.3 | 59.7 | 64.7 | 68.2 | 65.2 |
| Individual | 92.9 | 90.9 | 91.4 | 94.7 | 92.8 | 92.8 | 95.9 | 94.3 | 94.8 |
| *(Task-known scenarios)* | | | | | | | | | |
| Task Arithmetic (Ilharco et al., 2023) | $70.8_{(76.5)}$ | $65.3_{(72.1)}$ | $60.5_{(66.8)}$ | $75.4_{(79.6)}$ | $70.5_{(75.9)}$ | $65.8_{(70.8)}$ | $84.9_{(88.7)}$ | $79.4_{(84.0)}$ | $74.0_{(78.1)}$ |
| TIES (Yadav et al., 2023) | $75.1_{(81.0)}$ | $68.0_{(74.8)}$ | $63.4_{(69.9)}$ | $79.7_{(84.3)}$ | $73.2_{(78.7)}$ | $68.2_{(73.3)}$ | $86.9_{(90.7)}$ | $79.5_{(84.1)}$ | $75.7_{(79.8)}$ |
| Consensus TA (Wang et al., 2024b) | $75.0_{(80.8)}$ | $70.4_{(77.4)}$ | $65.4_{(72.0)}$ | $79.4_{(83.9)}$ | $74.4_{(79.9)}$ | $69.8_{(74.9)}$ | $86.3_{(90.1)}$ | $82.2_{(86.9)}$ | $79.0_{(83.2)}$ |
| EMR-Merging (Huang et al., 2024) | $91.3_{(97.8)}$ | $87.6_{(96.3)}$ | $87.4_{(95.6)}$ | $93.3_{(98.5)}$ | $90.5_{(97.6)}$ | $90.2_{(97.2)}$ | $95.2_{(99.3)}$ | $92.7_{(98.3)}$ | $92.8_{(97.9)}$ |
| **RETEX (Ours)** | $\mathbf{92.6}_{(99.7)}$ | $\mathbf{90.6}_{(99.6)}$ | $\mathbf{91.1}_{(99.7)}$ | $\mathbf{94.4}_{(99.7)}$ | $\mathbf{92.5}_{(99.7)}$ | $\mathbf{92.9}_{(99.6)}$ | $\mathbf{95.6}_{(99.7)}$ | $\mathbf{93.9}_{(99.6)}$ | $\mathbf{94.6}_{(99.8)}$ |
| *(Task-unknown scenarios)* | | | | | | | | | |
| Weight Averaging | $66.3_{(72.1)}$ | $64.3_{(71.1)}$ | $61.0_{(67.5)}$ | $72.2_{(76.6)}$ | $69.5_{(74.8)}$ | $65.3_{(70.4)}$ | $79.6_{(83.2)}$ | $76.7_{(81.1)}$ | $71.6_{(75.6)}$ |
| Twin-Merging (Lu et al., 2024) | $84.0_{(90.3)}$ | $70.0_{(76.7)}$ | $57.5_{(61.8)}$ | $91.4_{(96.2)}$ | $78.4_{(83.9)}$ | $63.1_{(67.0)}$ | $93.7_{(97.7)}$ | $86.2_{(91.2)}$ | $74.8_{(78.6)}$ |
| DaWin (Oh et al., 2025) | $89.0_{(95.3)}$ | $73.8_{(80.3)}$ | $52.8_{(57.7)}$ | $87.1_{(91.9)}$ | $77.8_{(83.5)}$ | $62.8_{(67.3)}$ | $91.6_{(95.5)}$ | $82.6_{(87.2)}$ | $77.5_{(81.8)}$ |
| **RETEX (Ours)** | $\mathbf{92.0}_{(99.1)}$ | $\mathbf{89.8}_{(98.8)}$ | $\mathbf{89.4}_{(97.9)}$ | $\mathbf{94.0}_{(99.2)}$ | $\mathbf{91.9}_{(99.1)}$ | $\mathbf{91.3}_{(98.0)}$ | $\mathbf{95.2}_{(99.4)}$ | $\mathbf{93.5}_{(99.2)}$ | $\mathbf{93.1}_{(98.3)}$ |

## 5.1 VISION TASKS

### 5.1.1 MERGING 8, 14, AND 20 VISION TASKS

**Setting.** For evaluating RETEX on varying scale vision tasks, we follow the setting of TALL-Mask (Wang et al., 2024b). We fine-tune a separate model for each dataset on three CLIP (Radford et al., 2021) backbones (ViT-B/32, ViT-B/16, and ViT-L/14) and then evaluate the multi-task merged model. The 8-task configuration comprises (a) SUN397 (Xiao et al., 2016), (b) Cars (Krause et al., 2013), (c) RESISC45 (Cheng et al., 2017), (d) EuroSAT (Helber et al., 2019), (e) SVHN (Netzer et al., 2011), (f) GTSRB (Stallkamp et al., 2011), (g) MNIST (Deng, 2012), and (h) DTD (Cimpoi et al., 2014). The 14-task setting extends this list with (i) CIFAR100 (Krizhevsky et al., 2009), (j) STL10 (Coates et al., 2011), (k) Flowers102 (Nilsback & Zisserman, 2008), (l) Oxford-IIIT-Pet (Parkhi et al., 2012), (m) PCAM (Veeling et al., 2018), and (n) FER2013 (Goodfellow et al., 2013). The 20-task setting further adds (o) EMNIST (Cohen et al., 2017), (p) CIFAR10 (Krizhevsky et al., 2009), (q) Food101 (Bossard et al., 2014), (r) FashionMNIST (Xiao et al., 2017), (s) RenderedSST2 (Socher et al., 2013), and (t) KMNIST (Clanuwat et al., 2018). Accuracy is reported as the evaluation metric for all datasets.

**Results.** Table 1 shows that RETEX achieves the best accuracy on all three CLIP backbones and for 8/14/20 tasks in both settings (task-known and task-unknown). Compared to strong task-known baselines such as EMR-Merging, RETEX yields consistent gains, and in the harder task-unknown setting it substantially surpasses Twin-Merging and DaWin across every backbone. Accuracy remains stable as the number of tasks increases: RETEX recovers at least 99.6% of individual-expert performance even at 20 tasks on every backbone. These results indicate that RETEX reliably reconstructs high-fidelity task experts from a single merged model while scaling to larger, more diverse task suites.

### 5.1.2 MERGING 30 VISION TASKS

**Setting.** To further assess scalability, we extend the ViT-B/32 evaluation to a challenging 30-task suite by augmenting the 20-task configuration (Section 5.1.1) with ten additional datasets: Vegetables (Ahmed et al., 2021), Kvasir-v2 (Pogorelov et al., 2017), Intel Images (Bansal, 2019), Weather (Xiao et al., 2021), Cats and dogs (Cukierski), MangoLeafBD (Ahmed et al., 2023), Beans (Lab, 2020), Landscape Recognition (DeepNets), Garbage Classification (CCHANG, 2018), and Fruits-360 (Muresan & Oltean, 2018). Following Task Arithmetic (Ilharco et al., 2023), each task-specific CLIP model (Radford et al., 2021) is fine-tuned for 2000 iterations with batch size 128 using AdamW (Loshchilov & Hutter, 2019; Kingma, 2015) (learning rate $1 \times 10^{-5}$, weight decay 0.1) and a cosine schedule with 200 warm-up steps.

**Results.** Table 2 shows that static baselines deteriorate as tasks increase (e.g., Weight Averaging reaches 59.1% with 64.2% normalized accuracy at 30 tasks), and input-dependent methods also drop (Twin-Merging to 60.1%, DaWin to 40.3%). In contrast, RETEX remains stable: in task-known it

Table 2: **Multi-task performance on ViT-B/32 across different numbers of vision tasks.** Values in parentheses $_{(\cdot)}$ indicate normalized accuracy (merged / individual). All evaluations use the ViT-B/32 backbone.

| Method | 8 tasks | 14 tasks | 20 tasks | 30 tasks |
|---|---|---|---|---|
| Zero-shot | 48.3 | 57.2 | 56.1 | 55.5 |
| Individual | 92.9 | 90.9 | 91.4 | 93.1 |
| *(Task-known scenarios)* | | | | |
| Task Arithmetic (Ilharco et al., 2023) | $70.8_{(76.5)}$ | $65.3_{(72.1)}$ | $60.5_{(66.8)}$ | $58.0_{(62.8)}$ |
| TIES (Yadav et al., 2023) | $75.1_{(81.0)}$ | $68.0_{(74.8)}$ | $63.4_{(69.9)}$ | $60.1_{(65.2)}$ |
| Consensus TA (Wang et al., 2024b) | $75.0_{(80.8)}$ | $70.4_{(77.4)}$ | $65.4_{(72.0)}$ | $63.4_{(68.5)}$ |
| EMR-Merging (Huang et al., 2024) | $91.3_{(97.8)}$ | $87.6_{(96.3)}$ | $87.4_{(95.6)}$ | $90.5_{(97.0)}$ |
| **RETEX (Ours)** | $\mathbf{92.6}_{(99.7)}$ | $\mathbf{90.6}_{(99.6)}$ | $\mathbf{91.1}_{(99.7)}$ | $\mathbf{92.9}_{(99.7)}$ |
| *(Task-unknown scenarios)* | | | | |
| Weight Averaging | $66.3_{(72.1)}$ | $64.3_{(71.1)}$ | $61.0_{(67.5)}$ | $59.1_{(64.2)}$ |
| Twin-Merging (Lu et al., 2024) | $84.0_{(90.3)}$ | $70.0_{(76.7)}$ | $57.5_{(61.8)}$ | $60.1_{(65.2)}$ |
| DaWin (Oh et al., 2025) | $89.0_{(95.3)}$ | $73.8_{(80.3)}$ | $52.8_{(57.7)}$ | $40.3_{(42.9)}$ |
| **RETEX (Ours)** | $\mathbf{92.0}_{(99.1)}$ | $\mathbf{89.8}_{(98.8)}$ | $\mathbf{89.4}_{(97.9)}$ | $\mathbf{92.3}_{(99.1)}$ |

Table 3: **Multi-task performance of merged RoBERTa-based models on eight GLUE datasets.** Bold values indicate the best performance among merging methods (excluding the individual experts).

| Method | Single-Sentence | | Similarity & Paraphrase | | | Inference | | | Avg. |
|---|---|---|---|---|---|---|---|---|---|
| | CoLA | SST2 | MRPC | STSB | QQP | MNLI | QNLI | RTE | |
| Individual | 0.6018 | 0.9404 | 0.8922 | 0.9063 | 0.9141 | 0.8720 | 0.9271 | 0.7906 | 0.8556 |
| *(Task-known scenarios)* | | | | | | | | | |
| Task Arithmetic (Ilharco et al., 2023) | 0.1878 | 0.8589 | 0.7990 | 0.7403 | 0.8378 | 0.5908 | 0.6967 | 0.6209 | 0.6665 |
| TIES (Yadav et al., 2023) | 0.2048 | 0.8440 | 0.8113 | 0.5819 | 0.8570 | 0.6465 | 0.7481 | 0.4296 | 0.6404 |
| EMR-Merging (Huang et al., 2024) | 0.3996 | 0.9335 | 0.8627 | 0.8277 | 0.8972 | 0.8545 | 0.8957 | 0.7437 | 0.8018 |
| **RETEX (Ours)** | **0.5919** | **0.9433** | **0.8880** | **0.8676** | **0.9117** | **0.8732** | **0.9248** | **0.7617** | **0.8453** |
| *(Task-unknown scenarios)* | | | | | | | | | |
| Weight Averaging | 0.1396 | 0.6411 | 0.6936 | 0.3184 | 0.7536 | 0.4219 | 0.5870 | 0.5523 | 0.5134 |
| Twin-Merging (Lu et al., 2024) | **0.6040** | 0.9410 | 0.8720 | **0.8640** | **0.9080** | 0.8190 | 0.9050 | **0.7740** | 0.8130 |
| DaWin (Oh et al., 2025) | 0.2447 | 0.9141 | 0.8566 | 0.6753 | 0.8671 | **0.8364** | 0.7968 | 0.6663 | 0.7322 |
| **RETEX (Ours)** | 0.5829 | **0.9449** | **0.8823** | 0.6984 | 0.9075 | 0.8179 | **0.9167** | 0.7653 | **0.8145** |

attains **92.9**% (**99.7**% normalized), and in task-unknown it reaches **92.3**% (**99.1**% normalized), closely matching its 8/14/20-task behavior.

## 5.2 NLP TASKS

**Setting.** For evaluating RETEX on NLP tasks our experimental setup aligns with established protocols from recent model merging studies. We utilize the RoBERTa(Liu et al., 2019) as the common pre-trained backbone from which individual task models are fine-tuned. Performance is assessed across eight diverse NLP benchmarks, consistent with prior work: SST-2 (Socher et al., 2013), MRPC (Dolan & Brockett, 2005), STS-B (Cer et al., 2017), QQP (Iyer et al., 2017), MNLI (Williams et al., 2017), QNLI (Rajpurkar et al., 2016), and RTE (Giampiccolo et al., 2007).

**Results.** To further validate the versatility of RETEX, we extend our evaluation to a suite of NLP tasks, complementing the aforementioned vision task experiments. The detailed performance metrics for these NLP benchmarks are presented in Table 3. The results clearly indicate that RETEX substantially outperforms existing model merging techniques when applied to language models. Notably, our approach recovers approximately 98.7% of the performance of the original task-specific fine-tuned models. While RETEX exhibits slightly lower performance than Twin-Merging specifically on the CoLA and RTE dataset, its average performance across all evaluated NLP tasks surpasses that of Twin-Merging by more than 3 percentage points, underscoring its overall effectiveness and robustness in the language domain.

## 5.3 COMPUTATIONAL COST

**Batch inference.** Although RETEX recovers an input-specific expert, it supports efficient batch inference by grouping samples that share the same predicted task ID. (i) We feed a minibatch of size

Table 4: **Inference cost with CLIP backbone (ViT-B/32).** We report the average performance and resource usage across all tasks in the 8 computer vision task scenario, assuming the task of the input sample is initially unknown.

| Method | Inference cost (per sample) | VRAM (GB) | Avg. performance |
|---|---|---|---|
| DaWin (Oh et al., 2025) | 0.63s | 5.5 | 89.0% |
| TWIN-Merging (Lu et al., 2024) | 0.03s | 5.6 | 84.0% |
| **RETEX (Sample-wise)** | 0.04s | **3.1** | **92.0%** |
| **RETEX (Group: $B = 64$)** | **0.005s** | 3.3 | **92.0%** |

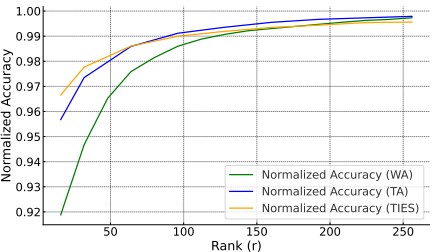

Figure 3: Normalized Accuracy across different ranks $r$, evaluated using three base merged models with RETEX: Weight Averaging (WA), Task Arithmetic (TA), and TIES-Merging (TIES).

Figure 4: Normalized accuracy as a function of task embedding dimension $d_{\text{emb}}$ for 8, 14, and 20 tasks. Results use a ViT-B/32 backbone with fixed recovery rank $r=256$ and Weight Averaging for $\theta_{\text{merge}}$.

$B$ through the task identifier router $\mathcal{R}_\phi$ that consumes the final embedding from the merged model and obtain predicted task IDs $\{\hat{t}_i\}_{i=1}^{B}$. (ii) We partition indices by task ID, $\mathcal{I}_t = \{i \in \{1, \ldots, B\} : \hat{t}_i = t\}$, and for each task ID $t$ with $|\mathcal{I}_t| > 0$ we generate once to form a group recovered model for $t$. (iii) We run the sub batches $X_{\mathcal{I}_t}$ in parallel through the corresponding group recovered model and then scatter outputs back to the original order. This procedure reduces the number of recoveries to $K$ unique task IDs in the batch with $K \leq \min(B, T)$, independent of batch size $B$, while adding only a single router pass and reusing one merged backbone and the lightweight projection module across groups.

**Results.** Table 4 shows that grouping by predicted task ID amortizes the recovery overhead across a minibatch. Concretely, RETEX reduces per-sample latency from $0.04$ s in sample-wise execution to $0.005$ s with group execution at $B=64$, while maintaining accuracy at $92.0\%$ and with only a small VRAM change (from $3.1$ GB to $3.3$ GB). Under the same task-unknown setting, grouped RETEX is both faster and more accurate than input-dependent baselines, outperforming Twin-Merging and DaWin. These gains arise because the number of recoveries scales with the number of unique task IDs in a batch, $K \leq \min(B, T)$, so a single recovered model per task ID serves its entire sub-batch ($K \ll B$ in practice).

## 5.4 ABLATION STUDY

We conduct ablation studies to analyze key hyperparameters in RETEX. Unless otherwise specified, all ablations follow the TALL-Mask (Wang et al., 2024b) setting on ViT-B/32.

**Other merged models.** To further investigate the generalization capability and broader applicability of RETEX, we extend its application beyond the default Weight Averaging (WA) base model. Specifically, we apply RETEX to merged models generated by Task Arithmetic (Ilharco et al., 2023) (TA) and TIES-Merging (Yadav et al., 2023) (TIES), where the merging coefficients specific to TA and TIES are kept fixed during the training of RETEX. As illustrated in Figure 3, utilizing a base merged model with inherently better performance (i.e., TA or TIES instead of WA) can lead to further, albeit modest, improvements in the final recovered accuracy achieved by RETEX. This performance advantage from using a more advanced base model is more pronounced at lower recovery ranks $r$. Nevertheless, these findings highlight a key strength of RETEX: its adaptability to integrate with various existing static merging techniques, potentially leveraging their individual strengths to further enhance multi-task performance.

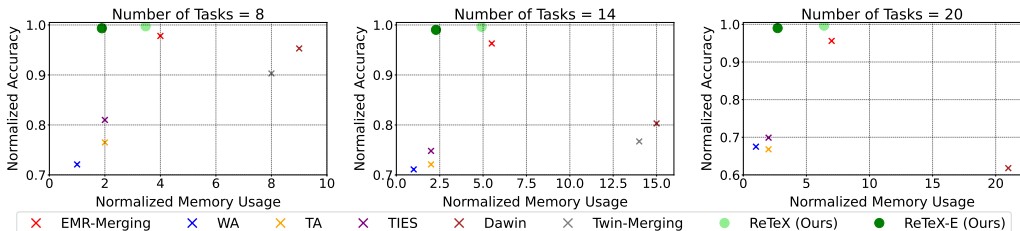

Figure 6: Comparison of normalized accuracy vs. normalized memory across model-merging methods, including RETEX (ReTeX) and RETEX-E (ReTeX-E). RETEX-E exhibits the strongest trade-off—lower memory at comparable or better accuracy—across different task counts.

**Task embedding dimension.** We investigate the influence of the task embedding dimension, $d_{\text{emb}}$, on the recovery performance of RETEX. For this analysis, we fix the rank $r=256$ and vary $d_{\text{emb}}$ while evaluating on configurations with 8, 14, and 20 tasks. Figure 4 shows that as $d_{\text{emb}}$ increases, the normalized accuracy generally improves and then saturates. Notably, even as the total number of tasks ($T$) increases, high recovery performance (approaching or exceeding 99.7%) can be achieved once $d_{\text{emb}}$ is sufficiently large, typically at or modestly above $T$. We further note that $d_{\text{emb}}$ can be determined relative to the number of tasks being merged without a significant performance loss.

## 5.5 ADVANCED EFFICIENCY

**Low-rank generation for task-adaptive components.**
Our default offset uses $\boldsymbol{\beta}_{\hat{t}}^{(l)} = \boldsymbol{\beta}_g^{(l)} \boldsymbol{\beta}_s^{(l)}$, where the task adaptation layer outputs $\boldsymbol{\beta}_g^{(l)} \in \mathbb{R}^{a \times r}$ and $\boldsymbol{\beta}_s^{(l)} \in \mathbb{R}^{r \times b}$ is a shared learnable factor. The dominant parameter cost comes from producing $\boldsymbol{\beta}_g^{(l)}$. To reduce it, we introduce **RETEX-E**, which factorizes the generator itself: $\boldsymbol{\beta}_g^{(l)} = \boldsymbol{\beta}_{gA}^{(l)} \boldsymbol{\beta}_{gB}^{(l)}$ with $\boldsymbol{\beta}_{gA}^{(l)} \in \mathbb{R}^{a \times r_g}$, $\boldsymbol{\beta}_{gB}^{(l)} \in \mathbb{R}^{r_g \times r}$, and $r_g < r$, yielding the final offset $\boldsymbol{\beta}_{\hat{t}}^{(l)} = (\boldsymbol{\beta}_{gA}^{(l)} \boldsymbol{\beta}_{gB}^{(l)}) \boldsymbol{\beta}_s^{(l)}$.

**Results.** Figure 5 evaluates the performance–memory trade-off by fixing the outer rank $r$ and varying the internal rank $r_g$. RETEX-E consistently matches or slightly exceeds the direct generator (RETEX) while using less

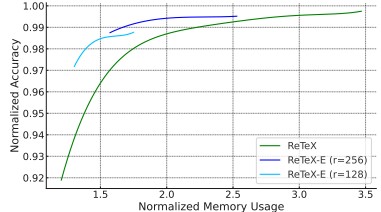

Figure 5: Normalized accuracy vs. normalized memory for RETEX (direct generator) and RETEX-E (two-stage generator) at fixed outer rank $r$ while varying $r_g$. RETEX-E attains a better memory–performance trade-off and maintains $>$ 99% recovery.

memory, especially in the high-accuracy regime, and preserves over 99% recovery. Complementarily, Figure 6 compares normalized accuracy vs. normalized memory across a wider set of merging baselines. Both RETEX and RETEX-E occupy the favorable top-left region, and RETEX-E in particular achieves an excellent trade-off, delivering $>$ 99% recovery with noticeably lower memory than alternatives.

## 6 CONCLUSION

In this work, we aim to recover task-expert-level performance while reducing memory usage overhead. We note that the task-specific offset between the task-expert parameters and the merged model parameters can be recovered from the merged model. Building upon this, we introduce a new model merging approach that learns to **Re**cover **T**ask e**X**perts (**RETEX**) from a merged model by predicting these offsets. Particularly, our framework first estimates the task identity for a given input. Conditioned on the estimated task identity, our framework generates a low-rank, task-dependent offset that maps the merged parameters to the corresponding expert for that input. Experimental results across vision and NLP domains highlight the effectiveness of RETEX in recovering task-expert-level performance while reducing memory overhead, compared to previous input-adaptive merging methods. We hope that this work encourages future research on the relationship between a merged model and task-specific models, and on more efficient approaches to model merging via offset recovery.

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

# A  JUSTIFICATION FOR OFFSET-BASED EXPERT RECOVERY

## A.1  AFFINE TRANSFORMATION−BASED TASK VECTOR RECOVERY

**Math derivation.**  Starting from the task–arithmetic formulation in the main paper,

$$\boldsymbol{\theta}_{\text{merge}} = \boldsymbol{\theta}_0 + \sum_{t=1}^{T} \lambda_t \boldsymbol{\tau}_t, \tag{7}$$

define the merged task vector $\boldsymbol{\tau}_{\text{merge}} \equiv \boldsymbol{\theta}_{\text{merge}} - \boldsymbol{\theta}_0$ and rewrite:

$$\boldsymbol{\tau}_{\text{merge}} = \sum_{i=1}^{T} \lambda_i \boldsymbol{\tau}_i = \lambda_t \boldsymbol{\tau}_t + \sum_{i \neq t} \lambda_i \boldsymbol{\tau}_i. \tag{8}$$

Isolating the target task $t$ by subtracting the non-$t$ terms from both sides gives

$$\boldsymbol{\tau}_{\text{merge}} - \sum_{i \neq t} \lambda_i \boldsymbol{\tau}_i = \lambda_t \boldsymbol{\tau}_t. \tag{9}$$

Assuming $\lambda_t$ is invertible (a nonzero scalar or an invertible linear operator), left–multiplying by $\lambda_t^{-1}$ yields

$$\lambda_t^{-1}\Big(\boldsymbol{\tau}_{\text{merge}} - \sum_{i \neq t} \lambda_i \boldsymbol{\tau}_i\Big) = \boldsymbol{\tau}_t, \tag{10}$$

$$\boldsymbol{\tau}_t = \lambda_t^{-1} \boldsymbol{\tau}_{\text{merge}} - \lambda_t^{-1} \sum_{i \neq t} \lambda_i \boldsymbol{\tau}_i. \tag{11}$$

Equation 11 shows that recovering $\boldsymbol{\tau}_t$ from $\boldsymbol{\tau}_{\text{merge}}$ involves a multiplicative term and an additive correction that compensates for interference from other tasks. This motivates the affine approximation

$$\boldsymbol{\tau}_t \approx \gamma_t \boldsymbol{\tau}_{\text{merge}} + \boldsymbol{\beta}_t, \tag{12}$$

where $\gamma_t$ (scalar) approximates $\lambda_t^{-1}$ and $\boldsymbol{\beta}_t$ (tensor) approximates $-\lambda_t^{-1} \sum_{i \neq t} \lambda_i \boldsymbol{\tau}_i$. If $\gamma_t, \boldsymbol{\beta}_t$ are generated from the task ID $t$, one can recover $\boldsymbol{\tau}_t$ from $\boldsymbol{\tau}_{\text{merge}}$ via Equation 12.

## A.2  LEARNING AND SIMPLIFYING THE AFFINE RULE IN RETEX

**Learning the full affine rule.**  Guided by Equation 12, RETEX generates the per–layer scalar $\gamma_t^{(l)}$ and a low–rank shift $\boldsymbol{\beta}_{t,g}^{(l)}$, composed with a shared factor $\boldsymbol{\beta}_s^{(l)}$:

$$\hat{\boldsymbol{\theta}}_t^{(l)} = \boldsymbol{\theta}_0^{(l)} + \gamma_t^{(l)}\big(\boldsymbol{\theta}_{\text{merge}}^{(l)} - \boldsymbol{\theta}_0^{(l)}\big) + \boldsymbol{\beta}_{t,g}^{(l)} \boldsymbol{\beta}_s^{(l)}. \tag{13}$$

**Observed behavior of the scaling factor.**  In practice, the task–averaged $\gamma_t^{(l)}$ for many layer types converges close to 1, indicating that most task–specific adjustment is carried by the shift term. Figure 7 visualizes this convergence trend over training.

**Impact of fixed scaling on performance.**  Motivated by the above observation, we fix $\gamma_t^{(l)} = 1$ for all tasks and layers so that RETEX generates only the low–rank shift:

$$\hat{\boldsymbol{\theta}}_t^{(l)} = \boldsymbol{\theta}_{\text{merge}}^{(l)} + \boldsymbol{\beta}_{t,g}^{(l)} \boldsymbol{\beta}_s^{(l)}. \tag{14}$$

Figure 8 shows that this fixed–scaling variant closely matches the accuracy of the learned–$\gamma$ model across a wide range of ranks $r$, while reducing memory usage. This supports the offset–only recovery perspective: recovering a task expert amounts to predicting a task–dependent offset from the merged parameters.

Equation 13 and Equation 14, together with the convergence in Figure 7, justify an offset–based expert recovery rule. Setting $\gamma_t^{(l)} = 1$ enables deployment with only a single merged model $\boldsymbol{\theta}_{\text{merge}}$; RETEX learns task–conditioned low–rank offsets $\boldsymbol{\beta}_{t,g}^{(l)} \boldsymbol{\beta}_s^{(l)}$ that effectively map $\boldsymbol{\theta}_{\text{merge}}$ to $\boldsymbol{\theta}_t$ without storing task experts.

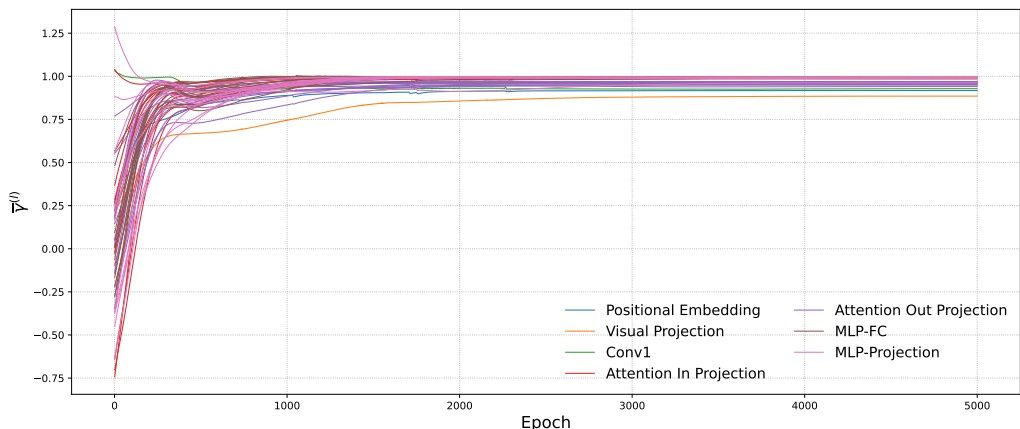

Figure 7: Convergence behavior of task-averaged $\gamma_t^{(l)}$ values for various 2D layer types during RETEX training (where $\gamma_t^{(l)}$ is learnable). Experiment on ViT-B/32 with 8 vision tasks shows many layers converge to $\gamma_t^{(l)} \approx 1$.

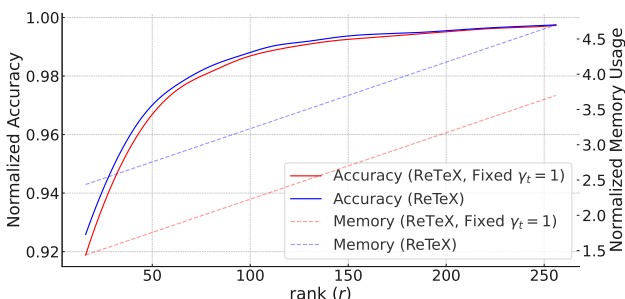

Figure 8: Performance and memory versus rank $r$ on 8 vision tasks (ViT-B/32). Fixing $\gamma_t^{(l)} = 1$ preserves accuracy relative to the learned–$\gamma$ model while lowering memory, validating the offset–only recovery view.

## B EXPERIMENT DETAILS

### B.1 MODULE ARCHITECTURE AND PARAMETERS

This section details the module architecture and parameterization of RETEX in the offset-only configuration justified in Appendix A. In this setting, RETEX learns to generate a task-specific offset tensor $\boldsymbol{\beta}_t^{(l)}$ for each layer $l$ and applies it to the merged parameters. The offset is factorized as $\boldsymbol{\beta}_t^{(l)} = \boldsymbol{\beta}_{t,g}^{(l)} \boldsymbol{\beta}_s^{(l)}$, where $\boldsymbol{\beta}_{t,g}^{(l)}$ is generated by a lightweight task adaptation MLP $h^{(l)}$ conditioned on a task embedding $e_t$, and $\boldsymbol{\beta}_s^{(l)}$ is a shared learnable component.

The core components involved in offset generation, their shapes (exemplified for a 2D layer), and their PyTorch-like forms are summarized in Table 5. We denote the number of tasks as $T$, the task embedding dimension as $d_{\text{emb}}$, the dimensions of a 2D layer's parameter matrix as $(a, b)$, and the chosen low-rank dimension as $r$.

The task adaptation MLP $h^{(l)}$ takes the $d_{\text{emb}}$-dimensional task embedding $e_t$ as input and generates the task-dependent factor $\boldsymbol{\beta}_{t,g}^{(l)}$. For a 2D layer of shape $(a, b)$, $\boldsymbol{\beta}_{t,g}^{(l)}$ has dimensions $(a, r)$, and the MLP output dimension is $a \cdot r$. The shared component $\boldsymbol{\beta}_s^{(l)}$ has dimensions $(r, b)$ and is a learnable parameter initialized with zeros. The final offset $\boldsymbol{\beta}_t^{(l)}$ is their product.

Table 5: **Core components for offset generation in RETEX (offset-only), exemplified for a 2D layer.** $T$: tasks, $d_{\text{emb}}$: task embedding dimension, $(a, b)$: 2D layer shape, $r$: rank.

| Component | Shape (for 2D layer) | Form |
|---|---|---|
| Task embedding $(e_t)$ | $(T, d_{\text{emb}})$ | nn.Embedding$(T, d_{\text{emb}})$ |
| Task adaptation MLP $(h^{(l)})$ | $(d_{\text{emb}}, a \cdot r)$ | nn.Linear$(d_{\text{emb}}, a \cdot r)$ |
| $\hookrightarrow$ Generated $\boldsymbol{\beta}_{t,g}^{(l)}$ | $(a, r)$ | Reshaped $h^{(l)}$ output |
| Shared $\boldsymbol{\beta}_s^{(l)}$ | $(r, b)$ | nn.Parameter(torch.zeros$(r, b)$) |
| Effective offset $\boldsymbol{\beta}_t^{(l)}$ | $(a, b)$ | $\boldsymbol{\beta}_t^{(l)} = \boldsymbol{\beta}_{t,g}^{(l)} \boldsymbol{\beta}_s^{(l)}$ |

**Parameter handling for diverse dimensions.** The shapes of the generated component $\boldsymbol{\beta}_{t,g}^{(l)}$ and the shared component $\boldsymbol{\beta}_s^{(l)}$ are adapted based on the dimensionality of the original layer parameter $\boldsymbol{\theta}^{(l)}$:

- **0D (Scalar) parameters:**
    - $\boldsymbol{\beta}_{t,g}^{(l)}$: Shape $(1)$, output of $h^{(l)}$.
    - $\boldsymbol{\beta}_s^{(l)}$: Shape $(1)$, learnable scalar.
- **1D (Vector) parameters:** For an original parameter of shape $(D)$:
    - $\boldsymbol{\beta}_{t,g}^{(l)}$: Shape $(D)$, output of $h^{(l)}$.
    - $\boldsymbol{\beta}_s^{(l)}$: Shape $(1)$, learnable scalar (scales $\boldsymbol{\beta}_{t,g}^{(l)}$ element-wise).
- **2D (Matrix) parameters:** For an original parameter of shape $(a, b)$ and using rank $r$:
    - $\boldsymbol{\beta}_{t,g}^{(l)}$: Shape $(a, r)$, output of $h^{(l)}$ (reshaped).
    - $\boldsymbol{\beta}_s^{(l)}$: Shape $(r, b)$, learnable matrix.
- **4D (Tensor, e.g., convolutional kernels) parameters:** For an original parameter of shape $(c_{out}, c_{in}, k_h, k_w)$, treat it as 2D by reshaping to $(c_{out}, c_{in} \cdot k_h \cdot k_w)$ for decomposition with rank $r$:
    - $\boldsymbol{\beta}_{t,g}^{(l)}$: Shape $(c_{out}, r)$, output of $h^{(l)}$ (reshaped).
    - $\boldsymbol{\beta}_s^{(l)}$: Shape $(r, c_{in} \cdot k_h \cdot k_w)$, learnable matrix.
    - The resulting $\boldsymbol{\beta}_t^{(l)}$ is reshaped back to $(c_{out}, c_{in}, k_h, k_w)$.

The task adaptation MLP $h^{(l)}$ adjusts its output dimensionality to produce the required $\boldsymbol{\beta}_{t,g}^{(l)}$ for each layer type.

**Rank adjustment.** RETEX uses a common target rank $r$ across layers. For 2D parameters of shape $(d_1, d_2)$ (or 4D parameters reshaped to such a 2D form), the rank is adjusted per layer: if $r \geq \min(d_1, d_2)$, the effective rank is set to $\lfloor \min(d_1, d_2)/2 \rfloor$; otherwise, the target rank $r$ is used. This ensures a practical low-rank structure for all layers.

## B.2 BASELINE DETAILS

The baseline approaches employed for comparative evaluation in our experiments are detailed as follows:

- **Individual Models**: This represents the standard performance benchmark where a distinct, fine-tuned model is dedicated to each specific task. These models operate independently and are not designed for multi-task execution.
- **Weight Averaging** (Shoemake, 1985; Utans, 1996): As a foundational technique in model merging, this method directly computes an average of the parameters from all constituent task-specific models. It operates under the simplifying assumption that all tasks contribute equally, hence applying uniform weighting to each model.

- **Task Arithmetic** (Ilharco et al., 2023): This approach first defines a "task vector" $\boldsymbol{\tau}_t$ for each task $t$ as the parametric difference between the fine-tuned model $\boldsymbol{\theta}_t$ and the initial pre-trained model $\boldsymbol{\theta}_0$ (i.e., $\boldsymbol{\tau}_t = \boldsymbol{\theta}_t - \boldsymbol{\theta}_0$). A unified model $\boldsymbol{\theta}_{\text{merge}}$ is then constructed by adding a scaled sum of these task vectors to the pre-trained parameters, formulated as $\boldsymbol{\theta}_{\text{merge}} = \boldsymbol{\theta}_0 + \lambda \sum_{t=1}^{T} \boldsymbol{\tau}_t$. The scaling factor $\lambda$ is a hyperparameter selected from the range $\{0.0, 0.1, \ldots, 1.0\}$ to maximize average performance across all task validation sets.

- **TIES-Merging** (Yadav et al., 2023): This method refines task vectors before merging through a three-step process: Trim, Elect Sign, and Merge. In the Trim step, only the top 20% of values by magnitude in each task vector are retained, with others zeroed out. The Elect Sign step (implicitly handled by the original task vector signs) and the subsequent Merge step proceed analogously to Task Arithmetic, including the hyperparameter tuning for the scaling factor.

- **Consensus TA** (Wang et al., 2024a): This technique first utilizes a multi-task model to derive binary masks that highlight parameters deemed critical for each task. The sparsity of these masks is controlled by a hyperparameter $\lambda$, optimized over $\{0.2, 0.3, 0.4, 0.5, 0.6\}$ using validation performance. Each task-specific mask is then applied to its corresponding task vector via an element-wise (Hadamard) product before the final merging, which follows the Task Arithmetic procedure.

- **EMR-Merging** (Huang et al., 2024): This approach begins by creating a consolidated "unified task vector" derived from the sign and magnitude of individual task vectors. It then computes task-specific binary masks and rescaling vectors for each task. The final merged model for a given task is obtained by an element-wise multiplication of this unified task vector with the corresponding task-specific mask and rescaler. This method is presented as hyperparameter-free.

- **Twin-Merging** (Lu et al., 2024): This method involves first training a router for dynamic task identification. A shared "common expert" is then established using Task Arithmetic with a predetermined scaling factor. Subsequently, "exclusive knowledge vectors" unique to each task are extracted, typically using Singular Value Decomposition (SVD) or a trimming procedure similar to TIES-Merging (with Trim reported as superior). At inference, the router assigns task-specific weights to an input. The final model output is derived by combining the shared expert with a weighted sum of these exclusive knowledge vectors, using the router-determined weights.

- **DaWin** (Oh et al., 2025): This dynamic merging technique assigns an input-specific weight to each task model. These weights are calculated based on the Shannon entropy of the outputs from both the task-specific model and the pre-trained base model for the given input. To optimize inference speed, a Beta Mixture Model (BMM) can optionally be trained to approximate these dynamic weights, typically using $K = 3$ mixture components by default.

### B.3 COMPUTATIONAL RESOURCES AND TRAINING TIME

All experimental procedures reported in this work, encompassing the training and inference of our proposed RETEX framework, as well as performance evaluations and computational cost measurements for baseline methods, were conducted on specific GPU hardware. For experiments involving 14 tasks or fewer, NVIDIA GeForce RTX 3090 GPUs were utilized. As a specific example of training duration, the training of RETEX for the 8-task vision benchmark typically completed in approximately 53 minutes and 49 seconds on a single NVIDIA GeForce RTX 3090 GPU. For more extensive experiments involving 20 tasks or more, NVIDIA H100 80GB HBM3 GPUs were employed to accommodate the increased computational demands.

## C MORE ABLATION STUDY

### C.1 RANK

We study how the low–rank dimension $r$ affects recovery quality and memory. Figure 9 reports normalized accuracy (mean accuracy divided by the corresponding individual-expert accuracy) and

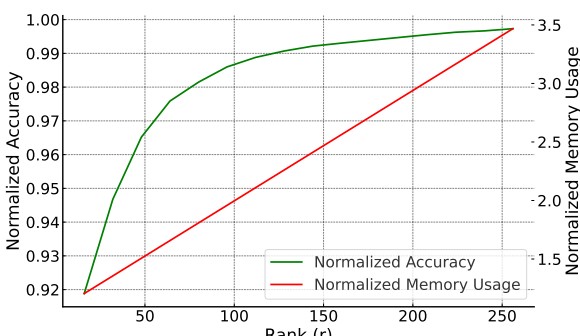

Figure 9: Normalized accuracy and required memory ratio as a function of rank $r$ on ViT-B/32. Accuracy crosses 99% at $r \geq 128$ and saturates near $r = 256$, which we adopt by default for subsequent experiments.

required memory ratio (memory relative to the base model) on ViT-B/32. Across a sweep of $r$, RETEX exceeds 99% normalized accuracy once $r \geq 128$, and the gains saturate beyond $r = 256$ (typically $\approx 99.7$–99.8%). Since the parameter and activation costs grow roughly linearly with $r$ through the factors $a \times r$ and $r \times b$, we set $r = 256$ in the main experiments to balance accuracy and memory.

## C.2 RANDOM SEED

Table 6: Normalized accuracy for each random seed and number of tasks across all tasks in the computer vision task with ViT-B/32 CLIP backbone. The bottom row reports the sample mean and its standard deviation (Mean $\pm$ std) over the five seeds.

| Seed | 8 tasks | 14 tasks | 20 tasks |
|---|---|---|---|
| 0 | 99.7352 | 99.6184 | 99.6430 |
| 1 | 99.7278 | 99.6201 | 99.6431 |
| 2 | 99.7364 | 99.6168 | 99.6431 |
| 3 | 99.7387 | 99.6159 | 99.6338 |
| 4 | 99.7317 | 99.6244 | 99.6467 |
| **Mean $\pm$ std** | $99.7340 \pm 0.0043$ | $99.6191 \pm 0.0034$ | $99.6419 \pm 0.0048$ |

To assess the stability and robustness of RETEX with respect to initialization and other sources of randomness in the training process, we conducted experiments across multiple random seeds. Table 6 presents the normalized accuracy of RETEX on the ViT-B/32 CLIP backbone for computer vision task suites of 8, 14, and 20 tasks, evaluated over five different random seeds (0 through 4). The results demonstrate a high degree of consistency across seeds. This low variance across different seeds indicates that the performance of RETEX is not highly sensitive to the specific random initialization used, suggesting reliable and reproducible outcomes.

## C.3 COSINE SIMILARITY AS AN OBJECTIVE

Prior works in model merging Yang et al. (2024); Huang et al. (2024); Xiong et al. (2024); Davari & Belilovsky (2024) have occasionally utilized cosine similarity as a metric, particularly to evaluate the alignment or proximity between different task vectors or between a task vector derived from a merged model and those from individual task-specific models. This metric captures the angular relationship between these vectors, providing insights into their directional agreement, which can be a complementary perspective to L2 distance that measures magnitude differences in the parameter space. Motivated by its use as an evaluative measure for task vector relationships, we investigate whether employing cosine similarity directly as the training objective for RETEX offers

any advantages or differing characteristics compared to our standard L2 reconstruction loss when reconstructing task-specific parameters.

Let $\mathcal{L}_2$ denote our original layer-wise L2 reconstruction loss as defined in the main paper Equation 4. For this ablation, we define a cosine similarity-based loss, $\mathcal{L}_{\cos}$, based on the overall task vectors. Let $\boldsymbol{\tau}_t = \boldsymbol{\theta}_t - \boldsymbol{\theta}_0$ be the target task vector for task $t$, representing the difference between the original fine-tuned model parameters $\boldsymbol{\theta}_t$ and the pre-trained model parameters $\boldsymbol{\theta}_0$. Similarly, let $\hat{\boldsymbol{\tau}}_t = \hat{\boldsymbol{\theta}}_t - \boldsymbol{\theta}_0$ be the reconstructed task vector, where $\hat{\boldsymbol{\theta}}_t$ are the parameters recovered by RETEX. For the purpose of calculating cosine similarity, $\boldsymbol{\tau}_t$ and $\hat{\boldsymbol{\tau}}_t$ are treated as single, high-dimensional vectors representing the entirety of these parameter differences. The cosine similarity loss $\mathcal{L}_{\cos}$ is then formulated as:

$$\mathcal{L}_{\cos} = \sum_{t=1}^{T} \left( 1 - \frac{\hat{\boldsymbol{\tau}}_t \cdot \boldsymbol{\tau}_t}{\|\hat{\boldsymbol{\tau}}_t\|_2 \cdot \|\boldsymbol{\tau}_t\|_2} \right) \tag{15}$$

where $\cdot$ represents the dot product between the (flattened) task vectors.

We conducted experiments to compare these objectives. As shown in Figure 10, the red dot indicates the normalized accuracy when using only $\mathcal{L}_{\cos}$ as the objective (effectively $\lambda = 0$ in a combined loss). The green dashed line represents the performance of our standard RETEX which uses only the $\mathcal{L}_2$ loss, without any cosine similarity component. Furthermore, we evaluated a combined objective function $\mathcal{L}_{\text{combined}} = \mathcal{L}_{\cos} + \lambda\mathcal{L}_2$, where $\lambda$ is a hyperparameter controlling the contribution of the L2 loss. The blue line in Figure 10 plots the normalized accuracy achieved with this combined loss for varying values of $\lambda$.

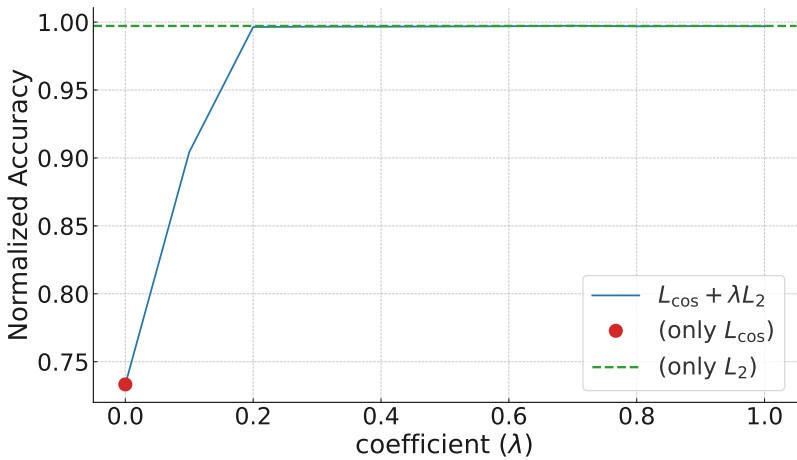

Figure 10: Normalized accuracy of RETEX when trained with different objective functions: only $\mathcal{L}_{\cos}$ (red dot, corresponding to $\lambda = 0$ in the combined loss), only $\mathcal{L}_2$ (green dashed line, our standard approach), and a combination $\mathcal{L}_{\cos} + \lambda\mathcal{L}_2$ (blue line) for various $\lambda$ coefficients. Experiments were conducted on the 8 vision task benchmark with the ViT-B/32 backbone.

**Analysis.** The results presented in Figure 10 demonstrate that using only cosine similarity ($\mathcal{L}_{\cos}$, red dot) as the training objective results in a normalized accuracy of approximately 0.73. This is substantially lower than the near-perfect recovery (normalized accuracy $\approx 1.00$) achieved when using only the L2 loss ($\mathcal{L}_2$), indicated by the green dashed line. This performance gap highlights that cosine similarity alone is insufficient for high-fidelity task expert recovery.

Interestingly, when combining the two losses as $\mathcal{L}_{\cos} + \lambda\mathcal{L}_2$, the performance (blue line) rapidly improves as the coefficient $\lambda$ for the L2 loss increases. Even with a relatively small $\lambda = 0.2$, the normalized accuracy of the combined loss already reaches the level achieved by the L2 loss alone (the green dashed line) and subsequently remains saturated at this high performance for $\lambda \geq 0.2$. This observation strongly suggests that the performance recovery is primarily driven by the L2 component of the loss. The fact that adding a small amount of $\mathcal{L}_2$ to $\mathcal{L}_{\cos}$ allows the model to match

the performance of $\mathcal{L}_2$ alone, and that further increasing the $\mathcal{L}_{\cos}$ component (by having smaller $\lambda$) does not improve beyond what $\mathcal{L}_2$ achieves, indicates that $\mathcal{L}_{\cos}$ offers little to no additional benefit for recovery when a sufficient L2 term is present.

Therefore, it can be inferred that the L2 distance is the main driver for effectively recovering the task experts. The improvement in cosine similarity (i.e., directional alignment) appears to be a natural consequence of minimizing the L2 distance between the reconstructed and target task vectors. If two vectors are made very close in Euclidean space (small L2 distance), their angular deviation will inherently decrease, leading to high cosine similarity. This suggests that directly optimizing for cosine similarity is not essential for, and may even distract from, the core objective of precise parameter reconstruction, for which L2 loss is more effective. Consequently, while cosine similarity can be an insightful metric, our standard L2 loss remains the more robust and primary objective function for RETEX.

### C.4 IMPACT OF FACTORIZATION ORDER IN SHIFT TENSOR GENERATION

In our proposed RETEX framework (hereafter referred to as RETEX for clarity in this comparison), the task-specific shift tensor $\boldsymbol{\beta}_t^{(l)}$ for a 2D layer of shape $(a, b)$ is generated via a low-rank factorization: $\boldsymbol{\beta}_t^{(l)} = \boldsymbol{\beta}_{t,g}^{(l)}\boldsymbol{\beta}_s^{(l)}$. Here, the task-adaptive component $\boldsymbol{\beta}_{t,g}^{(l)}$, generated by the Task Adaptation Layer ($h^{(l)}$), has dimensions $(a, r)$, and the shared component $\boldsymbol{\beta}_s^{(l)}$, a learnable parameter initialized with zeros, has dimensions $(r, b)$.

This ablation study investigates the impact of reversing the order of these factorized components. We explore an alternative formulation, denoted RETEX-Alt, where the shift tensor is constructed as $\boldsymbol{\beta}_t^{(l)} = \boldsymbol{\beta}_{s,\text{alt}}^{(l)}\boldsymbol{\beta}_{t,g,\text{alt}}^{(l)}$. In this alternative setup:

- $\boldsymbol{\beta}_{s,\text{alt}}^{(l)}$ is a shared, learnable parameter (initialized with zeros) with dimensions $(a, r)$.

- $\boldsymbol{\beta}_{t,g,\text{alt}}^{(l)}$ is the task-adaptive component generated by the Task Adaptation Layer, now with dimensions $(r, b)$.

The core idea is to examine whether making the first factor shared (and learnable from zero-initialization) and the second factor task-adaptive (generated by the Task Adaptation Layer) influences the model's recovery performance and memory usage, compared to our standard RETEX approach where the first factor is task-adaptive and the second is shared.

The experimental settings, including the ViT-B/32 backbone, 8 vision tasks, the training procedure for the Task Adaptation Layer, and varying rank $r$, remain consistent with our main experiments. The only modification is this reordering of the factorization for $\boldsymbol{\beta}_t^{(l)}$ and the corresponding change in the output shape of the Task Adaptation Layer.

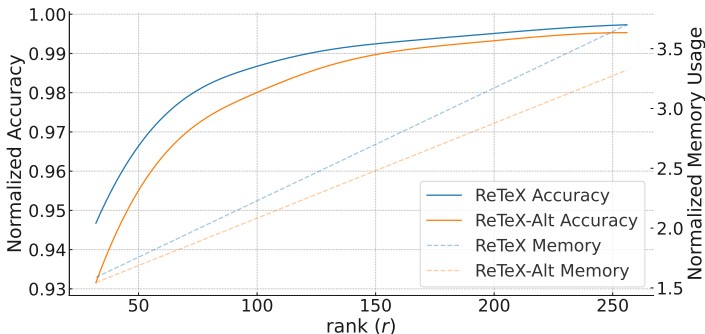

Figure 11: Comparison of normalized accuracy and normalized memory usage versus rank ($r$) for the standard ReTeX (blue lines) and ReTeX-Alt (orange lines, representing the alternative factorization order). Solid lines indicate accuracy, and dashed lines indicate memory usage. Experiments were conducted on 8 vision tasks with ViT-B/32.

**Analysis.** Figure 11 illustrates the normalized accuracy and normalized memory usage for both ReTeX (blue lines) and RETEX-Alt (orange lines) across different ranks $r$.

A key observation is that RETEX-Alt (orange dashed line) generally exhibits lower memory usage compared to RETEX (light blue dashed line) for the same rank $r$. This is primarily because in many neural network layers, the input dimension $a$ is often larger than the output dimension $b$ (i.e., $a > b$). In RETEX-Alt, the shared parameter $\boldsymbol{\beta}_{s,\mathrm{alt}}^{(l)}$ has shape $(a, r)$, while the Task Adaptation Layer generates $\boldsymbol{\beta}_{t,g,\mathrm{alt}}^{(l)}$ of shape $(r, b)$. Conversely, in RETEX, the Task Adaptation Layer generates $\boldsymbol{\beta}_{t,g}^{(l)}$ of shape $(a, r)$. Since the parameters of the Task Adaptation Layer contribute significantly to the overall memory, generating a smaller matrix (typically $(r, b)$ in RETEX-Alt when $b < a$) results in lower memory for RETEX-Alt.

However, this reduction in memory usage for RETEX-Alt is accompanied by a noticeable decrease in normalized accuracy (orange solid line) compared to the standard RETEX (blue solid line) across all ranks. For instance, at rank $r = 256$, RETEX achieves a normalized accuracy close to 1.00, while RETEX-Alt is visibly lower.

When comparing at roughly equivalent memory usage levels (e.g., by selecting a higher rank for RETEX-Alt to match the memory of RETEX at a lower rank, or vice-versa, though not directly shown on a single vertical line), the performance difference might appear less substantial. However, the consistent trend shows that for any given rank, RETEX slightly outperforms RETEX-Alt. This suggests that while reversing the factorization order can lead to memory savings due to typical layer dimensionalities, it may compromise the model's capacity to learn effective task-specific shifts. The standard RETEX configuration, where the larger task-adaptive component $\boldsymbol{\beta}_{t,g}^{(l)}$ (often $a \times r$ with $a > b$) is generated by the Task Adaptation Layer and then projected by the smaller shared, zero-initialized learnable parameter $\boldsymbol{\beta}_s^{(l)}$ ( $r \times b$), appears to offer a marginally better performance-to-rank trade-off. Placing the shared, zero-initialized learnable parameter as the second factor in the multiplication (as in the standard RETEX) might provide a more stable or effective learning dynamic for task-specific recovery. Therefore, our standard RETEX factorization order ($\boldsymbol{\beta}_{t,g}^{(l)}\boldsymbol{\beta}_s^{(l)}$) is retained as the primary approach.

## C.5   IMPACT OF CALIBRATION SAMPLE SIZE ON ROUTER PERFORMANCE

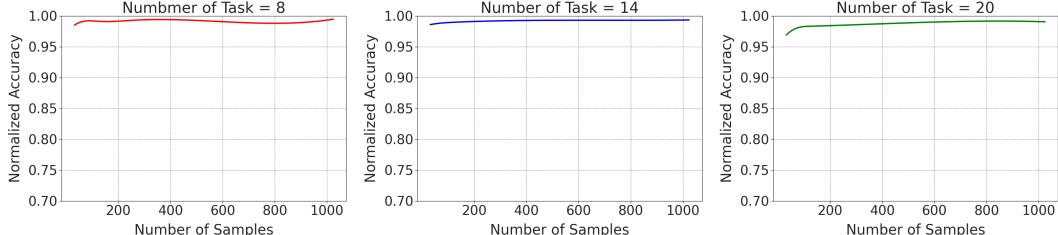

Figure 12: **Task router accuracy as a function of the number of training samples per task.** Results are shown for models trained on 8, 14, and 20 tasks with a ViT-B-32 backbone. The x-axis is on a log scale.

**Sample efficiency of the task router.**   A key design goal of our framework is efficiency, particularly in its data requirements for task identification. The task router ($\mathcal{R}_\phi$), as detailed in Section 4.2, is a lightweight network designed to operate effectively with a minimal number of calibration samples. To validate this sample efficiency, we conduct an ablation study on the number of samples per task used for training the router. As shown in Figure 13, we evaluate scenarios with 8, 14, and 20 tasks, varying the number of training samples from just 32 to 1024 per task.

The results highlight the practical utility of our approach. Even with as few as **32 samples** per task, the router achieves a remarkable normalized accuracy of **98.5%** for the 8-task scenario. The performance remains robust as the number of tasks increases, with the 20-task router achieving **96.9%** accuracy with the same minimal sample size. Moreover, accuracy exhibits a steady, gradual

improvement as the number of samples increases from 32 to 1024, indicating that while the router performs strongly with very few samples, its performance can be further refined with more data. This result confirms that the task router does not require large, task-specific datasets for calibration, which significantly reduces the data collection and training overhead associated with our framework. Consequently, this high sample efficiency makes our task classification mechanism a practical and scalable solution for multi-task scenarios.

### C.6 RETEX APPLICATOIN WITH SHARED LAYERS

RETEX recovers experts in a layer-wise manner because parameter interference induced by merging is not uniform across layers. Different blocks absorb and entangle task updates to varying degrees, so offsets must be tailored per layer to effectively undo these layer-specific deviations.

**RETEX-S (Shared layers).** To test whether sharing hurts recovery, we introduce RETEX-S: for all layers that share the same parameter shape, we tie the task adaptation layer $h^{(l)}$ and the shared factor $\boldsymbol{\beta}_s^{(l)}$ across those layers (i.e., a single generator and shared factor are reused for every layer in the shape group). All other components remain identical to RETEX.

**Results.** Figure 14 compares normalized accuracy and memory as the rank $r$ varies. Sharing across layers markedly degrades recovery quality: RETEX-S underperforms the layer-wise RETEX at virtually all ranks. Even when we equalize memory by increasing the rank of RETEX-S to match RETEX's parameter budget, RETEX-S still trails in accuracy, indicating that the drop is not merely a capacity issue but stems from forcing a single offset generator to explain heterogeneous, layer-dependent interference. These observations support the design choice to estimate offsets per layer rather than sharing them broadly.owever, this reduction in memory usage for RETEX-Alt is accompanied by a noticeable decrease in normalized accuracy (orange solid line) compared to the standard RETEX (blue solid line) across all ranks. For instance, at rank $r = 256$, RETEX achieves a normalized accuracy close to 1.00, while RETEX-Alt is visibly lower. When comparing at roughly equivalent memory usage levels (e.g., by selecting a higher rank for RETEX-Alt to match the memory of RETEX at a lower rank, or vice-versa, though not directly shown on a single vertical line), the performance difference might appear less substantial. However, the consistent trend shows that for any given rank, RETEX slightly outperforms RETEX-Alt. This suggests that while reversing the factorization order can lead to memory savings due to typical layer dimensionalities, it may compromise the model's capacity to learn effective task-specific shifts. The standard RETEX configuration, where the larger task-adaptive component $\boldsymbol{\beta}_{t,g}^{(l)}$ (often $a \times r$ with $a > b$) is generated by the Task Adaptation Layer and then projected by the smaller shared, zero-initialized learnable parameter $\boldsymbol{\beta}_s^{(l)}$ ($r \times b$), appears to offer a marginally better performance-to-rank trade-off. Placing the shared, zero-initialized learnable parameter as the second factor in the multiplication (as in the standard RETEX) might provide a more stable or effective learning dynamic for task-specific recovery. Therefore, our standard RETEX factorization order ($\boldsymbol{\beta}_{t,g}^{(l)}\boldsymbol{\beta}_s^{(l)}$) is retained as the primary approach.

### C.7 IMPACT OF CALIBRATION SAMPLE SIZE ON ROUTER PERFORMANCE

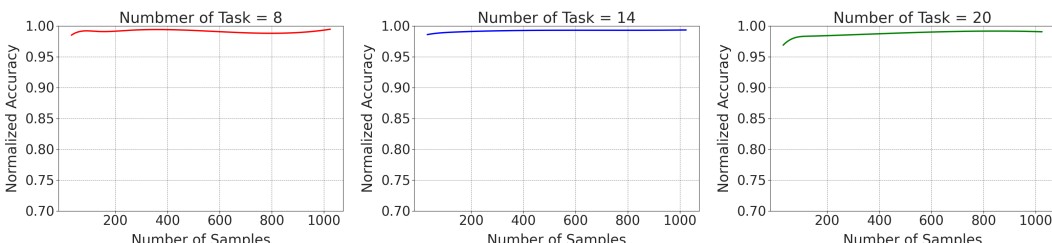

Figure 13: **Task router accuracy as a function of the number of training samples per task.** Results are shown for models trained on 8, 14, and 20 tasks with a ViT-B-32 backbone. The x-axis is on a log scale.

**Sample efficiency of the task router.** A key design goal of our framework is efficiency, particularly in its data requirements for task identification. The task router ($\mathcal{R}_\phi$), as detailed in Section 4.2, is a lightweight network designed to operate effectively with a minimal number of calibration samples. To validate this sample efficiency, we conduct an ablation study on the number of samples per task used for training the router. As shown in Figure 13, we evaluate scenarios with 8, 14, and 20 tasks, varying the number of training samples from just 32 to 1024 per task. The results highlight the practical utility of our approach. Even with as few as **32 samples** per task, the router achieves a remarkable normalized accuracy of **98.5%** for the 8-task scenario. The performance remains robust as the number of tasks increases, with the 20-task router achieving **96.9%** accuracy with the same minimal sample size. Moreover, accuracy exhibits a steady, gradual improvement as the number of samples increases from 32 to 1024, indicating that while the router performs strongly with very few samples, its performance can be further refined with more data. This result confirms that the task router does not require large, task-specific datasets for calibration, which significantly reduces the data collection and training overhead associated with our framework. Consequently, this high sample efficiency makes our task classification mechanism a practical and scalable solution for multi-task scenarios.

## C.8 RETEX APPLICATOIN WITH SHARED LAYERS

RETEX recovers experts in a layer-wise manner because parameter interference induced by merging is not uniform across layers. Different blocks absorb and entangle task updates to varying degrees, so offsets must be tailored per layer to effectively undo these layer-specific deviations.

## C.9 RETEX WITH SHARED LAYERS

RETEX recovers experts in a layer-wise manner because parameter interference introduced by merging is not uniform across layers. Different blocks absorb and entangle task updates to varying degrees, so offsets need to be tailored per layer to effectively undo these layer-specific deviations.

**RETEX-S (Shared layers).** To test whether sharing hurts recovery, we introduce RETEX-S: for all layers that share the same parameter shape, we tie the task adaptation layer $h^{(l)}$ and the shared factor $\beta_s^{(l)}$ across those layers (a single generator and shared factor are reused for every layer in the shape group). All other components remain identical to RETEX.

**Results.** Figure 14 compares normalized accuracy and memory as the rank $r$ varies. Sharing across layers markedly degrades recovery quality: RETEX-S underperforms the layer-wise RETEX at virtually all ranks. Even when memory is equalized by increasing the rank of RETEX-S to match RETEX's parameter budget, RETEX-S still trails in accuracy, indicating that the drop is not merely a capacity issue but stems from forcing a single offset generator to explain heterogeneous, layer-dependent interference. These observations support estimating offsets per layer rather than sharing them broadly.

# D ADDITIONAL EXPERIMENTS

## D.1 ROBUSTNESS TO UNSEEN TASK SCENARIOS

We utilize a lightweight router, $\mathcal{R}_\phi$, to select the appropriate expert model for an input $x$. The router processes a feature embedding from a unified model, $\theta_{\text{merge}}$, to produce a logit vector over $T$ known tasks. The predicted task, $\hat{t}$, is determined by the argmax operation (Equation 4.2). The router is trained with cross-entropy loss on a small calibration set:

$$\mathcal{L}_{\text{CE}} = -\frac{1}{N} \sum_{i=1}^{N} \sum_{t=1}^{T} y_{i,t} \log \left( \text{softmax} \left( \mathcal{R}_\phi \left( \theta_{\text{merge}}(x_i) \right) \right)_t \right) \tag{16}$$

where $N$ is the number of calibration samples, $y_{i,t}$ is the ground truth label (1 if sample $i$ belongs to task $t$, 0 otherwise). This enables the router to efficiently learn task boundaries with minimal memory overhead. Beyond task classification, we leverage the **entropy** of the router's output distribution

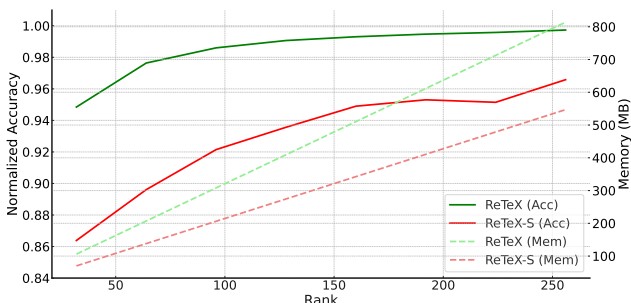

Figure 14: **Ablation: sharing generators across layers (RETEX-S).** Normalized accuracy vs. memory when tying the task adaptation layer and $\beta_s$ across shape-identical layers. RETEX-S consistently lags behind the default layer-wise RETEX, even under matched memory, suggesting that interference must be mitigated at the per-layer level rather than with a shared generator.

as a robust proxy for uncertainty. Our key observation is that in-distribution inputs yield low-entropy (confident) predictions, whereas unseen tasks result in high-entropy (uncertain) predictions.

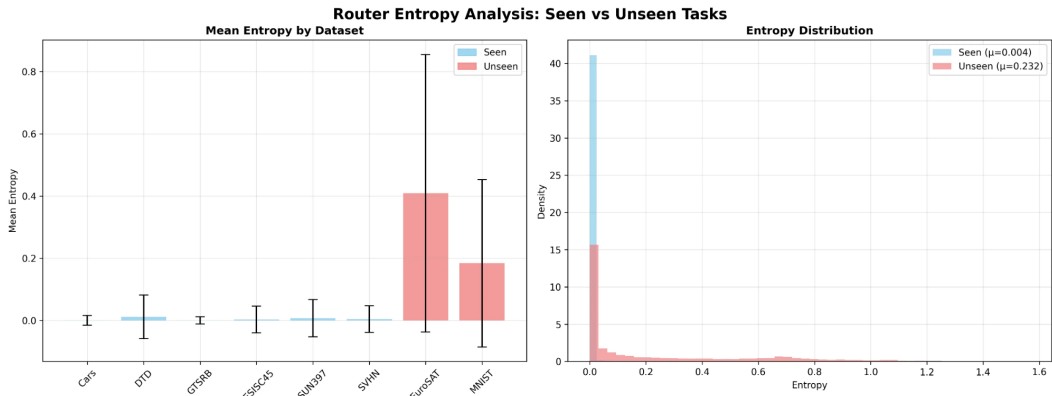

Figure 15: **Entropy-based OOD Detection.** Entropy distributions of router outputs on seen tasks (Cars, DTD, GTSRB, RESISC45, SUN397, SVHN) and unseen tasks (EuroSAT, MNIST) using a ViT-B/32 backbone. **(Left)** Entropy values clearly separate seen and unseen tasks. **(Right)** Aggregated distributions confirm a distinct gap, enabling threshold-based OOD detection.

Figure 15 further illustrates why this is possible: entropy distributions of seen tasks form a sharp low-entropy cluster, while unseen tasks produce clearly higher-entropy values. This separation enables robust OOD detection via a simple threshold, highlighting the generality and practicality of our approach.

Table 7 shows that our router maintains high Precision and Recall across different ViT backbones. This confirms that entropy-based OOD detection achieves strong and consistent classification performance regardless of the backbone architecture.

| Model | Accuracy | Recall | F1-Score |
|---|---|---|---|
| ViT-B/32 | 0.9670 | 0.9124 | 0.8893 |
| ViT-L/14 | 0.9641 | **0.9469** | 0.8846 |
| ViT-B/16 | **0.9701** | 0.9010 | **0.8975** |

Table 7: Accuracy, Recall, and F1-Score comparison across different ViT backbone models, demonstrating the robustness of entropy-based OOD detection across architectures.

Table 8: **Multi-task performance on GLUE with GPT-2 decoder models.** All rows (except the upper bound) are obtained by merging task experts fine-tuned on seven GLUE tasks (CoLA, MNLI, MRPC, QNLI, QQP, RTE, SST-2). Bold numbers indicate the best performance among merging methods, excluding the individual task experts.

| Method | CoLA | MNLI | MRPC | QNLI | QQP | RTE | SST-2 | Avg. |
|---|---|---|---|---|---|---|---|---|
| Individual | 76.8 | 82.1 | 80.4 | 88.3 | 89.6 | 65.3 | 91.2 | 82.0 |
| Weight Averaging | 55.0 | 55.1 | 51.0 | 57.6 | 76.7 | 44.8 | 52.5 | 56.1 |
| Fisher Merging (Matena, 2022) | 54.8 | 58.0 | 39.5 | 63.3 | 81.5 | 49.1 | 52.5 | 58.7 |
| RegMean (Jin et al., 2023b) | 61.7 | 70.4 | 65.4 | 69.7 | 78.8 | 56.0 | 79.7 | 68.8 |
| Task Arithmetic (Ilharco et al., 2023) | 68.7 | 68.6 | 69.6 | 70.5 | 81.8 | 47.3 | 83.6 | 70.0 |
| TIES-Merging (Yadav et al., 2023) | 68.4 | 71.4 | 68.4 | 69.6 | 82.4 | 47.7 | 81.8 | 70.0 |
| EMR-Merging (Huang et al., 2024) | 72.8 | 81.1 | 79.2 | 84.8 | 88.1 | **66.5** | 90.3 | 80.4 |
| RETEX (Ours) | **76.8** | **82.0** | 79.9 | **87.8** | **89.4** | 65.0 | **90.8** | **81.7** |

## D.2 Decoder-based NLP Tasks

**Settings.** To evaluate the recovery performance of RETEX, we follow the experimental setup of EMR-Merging Huang et al. (2024) and use GPT-2 Achiam et al. (2023) as a shared pre-trained backbone from which individual task models are fine-tuned. Performance is assessed across seven diverse NLP benchmarks, consistent with prior work: SST-2 Socher et al. (2013), MRPC Dolan & Brockett (2005), QQP Iyer et al. (2017), MNLI Williams et al. (2017), QNLI Rajpurkar et al. (2016), and RTE Giampiccolo et al. (2007).

**Results.** Table 8 shows that RETEX surpasses EMR-Merging by 1.3 points on average and outperforms other merging baselines by over 11 points. Compared with individual experts, the gap is only 0.3 points (99.6% retained), indicating strong recovery performance. These results mirror the trends observed with encoder backbones and support the applicability of RETEX to decoder-only language models.

## E Training-free Routing

We adopt a training-free, distributional classifier that uses intermediate features at a chosen layer. Fix a layer index $l \in \{1, \dots, L\}$ by validation. For each task $t$, collect a small calibration set and forward each sample through the corresponding task-specific fine-tuned model up to layer $l$ to obtain features $f_t^{(l)}(\boldsymbol{x})$. Fit a Gaussian $\mathcal{N}\big(\boldsymbol{\mu}_t^{(l)}, \Sigma_t^{(l)}\big)$ to these calibration features to model the task-$t$ distribution at layer $l$.

At test time, given a task-unknown input $\boldsymbol{x}$, forward $\boldsymbol{x}$ through each task-$t$ fine-tuned model up to layer $l$ to obtain $f_t^{(l)}(\boldsymbol{x})$, then compute the Mahalanobis distance to the corresponding task distribution:

$$\mathcal{M}_t^{(l)}(\boldsymbol{x}) = \big(f_t^{(l)}(\boldsymbol{x}) - \boldsymbol{\mu}_t^{(l)}\big)^\top \big(\Sigma_t^{(l)}\big)^{-1} \big(f_t^{(l)}(\boldsymbol{x}) - \boldsymbol{\mu}_t^{(l)}\big). \tag{17}$$

The predicted task ID is selected by

$$\hat{t} = \underset{t \in \{1, \dots, T\}}{\arg\min} \ \mathcal{M}_t^{(l)}(\boldsymbol{x}). \tag{18}$$

This formulation remains fully training-free: we estimate $\big\{\boldsymbol{\mu}_t^{(l)}, \Sigma_t^{(l)}\big\}_{t=1}^{T}$ from a small calibration set and require no learned router. In practice we instantiate routing with one validated layer $l$ for efficiency, but the approach is agnostic to the layer choice and can be applied at any layer without aggregating across layers.

**Experimental comparison.** We evaluate the proposed training-free routing in the task-unknown setting under the same experimental protocol as Section 5.1.1. As shown in Table 9, RETEX with training-free routing (RETEX-*Training free*) achieves accuracy very close to the learned-router variant (within ∼0.3–1.3 percentage points) across all CLIP backbones (ViT-B/32, ViT-B/16, ViT-L/14) and task counts (8/14/20). Despite requiring no router training, the training-free variant consistently

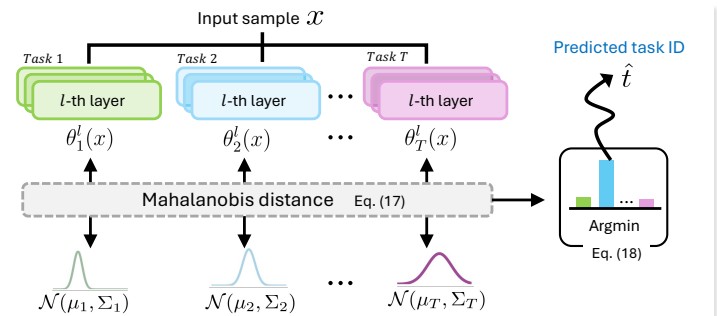

Figure 16: **Training-free Mahalanobis routing.** For a validated layer $l$, each task $t$ builds a Gaussian model $\mathcal{N}(\boldsymbol{\mu}_t^{(l)}, \Sigma_t^{(l)})$ from calibration features $f_t^{(l)}(\boldsymbol{x})$ extracted by forwarding samples through the task's fine-tuned model up to layer $l$. At test time, an input $\boldsymbol{x}$ is forwarded up to the same layer for every task-specific model, its Mahalanobis distance to each task distribution is computed ( Equation 17), and the predicted task is chosen by the $\arg\min$ over tasks ( Equation 18). The router requires no training, operates with a single chosen layer for efficiency, and is agnostic to the particular layer used.

Table 9: **Task-unknown multi-task merging results on vision benchmarks.** Top-1 accuracy (%) across CLIP backbones (ViT-B/32, ViT-B/16, ViT-L/14) and task counts (8/14/20) when task identity is not provided at inference.

| Method | ViT-B/32 | | | ViT-B/16 | | | ViT-L/14 | | |
|---|---|---|---|---|---|---|---|---|---|
| | 8 tasks | 14 tasks | 20 tasks | 8 tasks | 14 tasks | 20 tasks | 8 tasks | 14 tasks | 20 tasks |
| Zero-shot | 48.3 | 57.2 | 56.1 | 55.3 | 61.3 | 59.7 | 64.7 | 68.2 | 65.2 |
| Individual | 92.9 | 90.9 | 91.4 | 94.7 | 92.8 | 92.8 | 95.9 | 94.3 | 94.8 |
| Weight Averaging | $66.3_{(72.1)}$ | $64.3_{(71.1)}$ | $61.0_{(67.5)}$ | $72.2_{(76.6)}$ | $69.5_{(74.8)}$ | $65.3_{(70.4)}$ | $79.6_{(83.2)}$ | $76.7_{(81.1)}$ | $71.6_{(75.6)}$ |
| Twin-Merging (Lu et al., 2024) | $84.0_{(90.3)}$ | $70.0_{(76.7)}$ | $57.5_{(61.8)}$ | $91.4_{(96.2)}$ | $78.4_{(83.9)}$ | $63.1_{(67.0)}$ | $93.7_{(97.7)}$ | $86.2_{(91.2)}$ | $74.8_{(78.6)}$ |
| DaWin (Oh et al., 2025) | $89.0_{(95.3)}$ | $73.8_{(80.3)}$ | $52.8_{(57.7)}$ | $87.1_{(91.9)}$ | $77.8_{(83.5)}$ | $62.8_{(67.3)}$ | $91.6_{(95.5)}$ | $82.6_{(87.2)}$ | $77.5_{(81.8)}$ |
| **RETEX-*Training free* (Ours)** | $91.7_{(98.6)}$ | $88.6_{(97.4)}$ | $88.7_{(97.0)}$ | $93.1_{(98.2)}$ | $90.7_{(97.7)}$ | $90.9_{(97.5)}$ | $94.3_{(98.3)}$ | $92.5_{(98.1)}$ | $91.8_{(97.0)}$ |
| **RETEX (Ours)** | $\mathbf{92.0}_{(99.1)}$ | $\mathbf{89.8}_{(98.8)}$ | $\mathbf{89.4}_{(97.9)}$ | $\mathbf{94.0}_{(99.2)}$ | $\mathbf{91.9}_{(99.1)}$ | $\mathbf{91.3}_{(98.0)}$ | $\mathbf{95.2}_{(99.4)}$ | $\mathbf{93.5}_{(99.2)}$ | $\mathbf{93.1}_{(98.3)}$ |

surpasses prior input-dependent merging methods (Twin-Merging and DaWin) on every backbone and at every task scale, and maintains high recovery as the number of tasks increases. These results indicate that a calibration-only Mahalanobis routing scheme is sufficient to unlock most of the gains of RETEX in task-unknown scenarios while preserving strong scalability across architectures and task counts.

# F   LLM USAGE DISCLOSURE

We used a large language model solely to refine wording and improve clarity. The model did not contribute to research design, literature search, data generation or processing, coding, experimental analysis, or the production of technical content such as equations or proofs. All edits suggested by the model were reviewed and finalized by the authors, who accept full responsibility for the manuscript. The model is not an author; authorship, copyright, and research-ethics obligations rest entirely with the authors.

