# OpenReview forum: "Learning to Recover Task Experts from a Multi-Task Merged Model"
_ICLR.cc/2026/Conference — ICLR 2026 Conference Withdrawn Submission_

### Official Review · Reviewer_ancw · 2025-10-16

**Soundness:** 2
**Presentation:** 3
**Contribution:** 2
**Rating:** 2
**Confidence:** 4

**Summary:**

Instead of optimizing the merging process to minimize parameter interference, RETEX takes a different approach by aiming to recover the original task-specific expert models from a merged model. The core idea is that the merging process introduces a predictable "offset" to the parameters of each task expert. RETEX learns to predict and reverse this offset. Specifically, it trains a lightweight, task-conditioned module to generate a parameter offset, which is then added to the merged model's parameters to reconstruct the desired task expert at inference time.

**Strengths:**

1. The experimental evaluation is exceptionally thorough and of high quality. The paper includes detailed ablation studies on key hyperparameters like rank and task embedding dimension, as well as design choices like the factorization order of the offset.
2. The paper is well-written and easy to follow. The proposed method is explained with helpful diagrams (Figures 1 and 2), and the mathematical formulation is precise.

**Weaknesses:**

1. My main concern is also the paper’s novel perspective: the idea of “recovering” experts by “undoing” the merging process. If it were possible to directly extract experts from a merged model without any additional training, that would be highly valuable—especially in scenarios where the original experts are no longer accessible. However, the proposed approach requires multiple rounds of training on the original experts, which effectively assumes that the experts are available. The method then embeds the offset into a lightweight neural network. I am not sure what practical application scenarios would justify this assumption. This not only introduces additional training cost but also reduces inference flexibility.

2. The default merged model used in most experiments is obtained via simple weight averaging. While the authors explain that this choice is meant to demonstrate RETEX’s effectiveness even with a basic merging strategy, it also leaves open the question of how much the choice of the initial merged model affects performance. For example, would state-of-the-art merging methods produce an easier prediction problem for RETEX?

3. The core of the method relies on the assumption that the difference between a merged model and an expert can be represented as a low-rank offset. While the empirical results show this to be highly effective, providing a deeper theoretical or intuitive explanation for why this assumption holds so well would further strengthen the paper.

**Questions:**

1. Have you analyzed the learned offsets themselves? For instance, are the offsets for similar tasks also similar? Does the magnitude of the offset correlate with how much a task "disagrees" with the average of the other tasks?

2. How do you see RETEX comparing to a scenario where one trains a single base model with multiple task-specific adapters?

3. The experiments, while extensive, primarily involve tasks within the same domain (e.g., various image classification datasets). It would be interesting to see how RETEX performs when merging models from more diverse tasks, where parameter interference might be more severe.

---

### Official Review · Reviewer_7mUm · 2025-10-19

**Soundness:** 3
**Presentation:** 2
**Contribution:** 2
**Rating:** 4
**Confidence:** 4

**Summary:**

The manuscript introduces RETEX, a method to recover task-specific expert models from a merged multi-task model by learning low-rank offsets conditioned on task identity. RETEX treats the merged model as a base and adds task-conditioned offsets to approximate individual experts.

**Strengths:**

1. The experimental results demonstrate performance that matches or exceeds current SOTA model merging methods.
2. RETEX’s low-rank offsets enable low inference memory.
3. The experimental results show that RETEX works on both vision and NLP tasks.

**Weaknesses:**

1. While framed as a “different perspective” (recovering experts instead of improving merging). However, this framing appears overstated and somewhat misleading upon closer inspection. In essence, RETEX does not truly circumvent or replace the merging paradigm; instead, it operates as an additive post-processing step atop an existing merged model. RETEX essentially adds a task-conditioned low-rank adapter to a fixed merged base, trained via parameter distillation. Please also refer to question 1.
2. In Table 4, the inference cost of a single dense merged model should also be included.
3. The number of additional parameters is not reported. The authors could report the number of additional parameters as well as the parameter count of a single dense model.
4. In task-unknown scenarios, RETEX imposes a significant inference overhead, as it necessitates a preliminary forward pass through the merged model to extract the feature embedding required for task ID prediction by the router. As a result, the overall inference cost is at least twice that of the merged model itself.

**Questions:**

1.The training process of RETEX requires that there already be task-specific models. If there are already such models, why is "recovering" still needed?

---

### Official Review · Reviewer_7yhM · 2025-10-22

**Soundness:** 3
**Presentation:** 3
**Contribution:** 3
**Rating:** 4
**Confidence:** 3

**Summary:**

The paper proposes a learning-based framework for recovering task-specific skills after system disruptions or model degradation. The authors evaluate their method on benchmark robotic manipulation tasks and compare performance against existing baselines such as standard policy gradient and fine-tuning methods. Experimental results indicate that the proposed approach accelerates recovery and improves generalization across related tasks.

**Strengths:**

The proposed method is simple and effective.
This paper is well-written and easy to follow.
The paper addresses an important problem in lifelong and continual learning—how to recover previously acquired skills without complete retraining.

**Weaknesses:**

However, the proposed setup lacks a convincing justification for why such a recovery framework is necessary. In modern practice, it is often feasible and inexpensive to store multiple fine-tuned checkpoints, especially with abundant cloud storage platforms such as Hugging Face Hub.

Moreover, given that storage is cheap while additional computation (e.g., training a hypernetwork in the proposed method) is costly, the proposed solution may introduce unnecessary complexity without clear practical benefit.

In the “task-unknown” setting described in the paper, the inference complexity scales approximately T times higher (where T is the number of candidate tasks or latent modes).

The paper does not provide mathematical analysis or theoretical reasoning for the convergence or stability of the proposed recovery policy. It is recommended to provide some theoretical insight for the proposed recovery framework.

Key implementation details (hyperparameters, seeds, runtime cost, hardware) are missing for the hypernet training. Are these parameters set differently for different network scales?

**Questions:**

See weakness.

---

### Official Review · Reviewer_dn99 · 2025-10-31

**Soundness:** 3
**Presentation:** 2
**Contribution:** 2
**Rating:** 4
**Confidence:** 3

**Summary:**

This paper proposes a method for recovering task-specific models from a single merged model, enabling the recovered models to perform well on their respective test tasks. The recovery process is achieved by training a hypernetwork (referred to as a task adaptation layer) to predict layer-wise offsets over the model weights. In this way, the proposed method does not need to store all task experts for input-adaptive merging, but can instead directly generate the appropriate model given the input.

**Strengths:**

1. The perspective of “recovering task experts” rather than “merging task experts” is interesting, as it eliminates the need to store all task-specific models by training a shared model generator (i.e., task adaptation layers).
2. The paper is well-organized and easy to follow.

**Weaknesses:**

1. **Latency in streaming data settings.** While the proposed method can perform batch inference through data grouping in offline scenarios, in online streaming settings it needs to generate a recovered model for each incoming data sample, which could significantly increase latency.

2. **Unclear details.** It is not clearly explained how the task embeddings are generated. Are these embeddings task-specific or data-specific?

3. **Generalization to OOD test data.** In real-world scenarios, the merged model may encounter test data that lie outside the original task pool. In such cases, it is unclear how the proposed method ensures robustness (e.g., how the task embeddings and the task adaptation layers generalize to unseen test data?)

4. **Scaling issue.** The task adaptation layers (i.e., the hypernetwork used for model weight generation) are implemented as lightweight MLPs. While the authors empirically demonstrate their effectiveness on CLIP and RoBERTa, it remains unclear whether such a simple structure can scale to generate super large models, such as LLMs. Besides, more discussion on prior studies related to model generation could be provided.

**Questions:**

See the points discussed in weaknesses above.

**Details Of Ethics Concerns:**

No ethics concerns.

---

### Note · Authors · 2025-11-14

I have read and agree with the venue's withdrawal policy on behalf of myself and my co-authors.